# Qualitative Mechanism Independence

**Oliver E. Richardson**
Dept of Computer Science
Cornell University
Ithaca NY 14853
oli@cs.cornell.edu

**Spencer Peters**
Dept of Computer Science
Cornell University
Ithaca NY 14853
speters@cs.cornell.edu

**Joseph Y. Halpern**
Dept of Computer Science
Cornell University
Ithaca NY 14853
halpern@cs.cornell.edu

## Abstract

We define what it means for a joint probability distribution to be *(QIM-)compatible* with a set of independent causal mechanisms, at a qualitative level—or, more precisely, with a directed hypergraph $\mathcal{A}$, which is the qualitative structure of a probabilistic dependency graph (PDG). When $\mathcal{A}$ represents a qualitative Bayesian network, QIM-compatibility with $\mathcal{A}$ reduces to satisfying the appropriate conditional independencies. But giving semantics to hypergraphs using QIM-compatibility lets us do much more. For one thing, we can capture functional *dependencies*. For another, QIM-compatibility captures important aspects of causality: we can use compatibility to understand cyclic causal graphs, and to demonstrate compatibility is essentially to produce a causal model. Finally, compatibility has deep connections to information theory. Applying compatibility to cyclic structures helps to clarify a longstanding conceptual issue in information theory.

## 1 Introduction

The structure of a (standard) probabilistic graphical model (like a Bayesian Network or Markov Random Field) encodes a set of conditional independencies among variables. This is useful because it enables a compact description of probability distributions that have those independencies; it also lets us use graphs as a visual language for describing important qualitative properties of a probabilistic world. Yet these kinds of independencies are not the only important qualitative aspects of a probability measure. In this paper, we study a natural generalization of standard graphical model structures that can describe far more than conditional independence.

For example, another qualitative aspect of a probability distribution is that of functional *dependence*, which is also exploited across computer science to enable compact representations and simplify probabilistic analysis. Acyclic causal models, for instance, specify a distribution via a probability over *contexts* (the values of variables whose causes are viewed as outside the model), and a collection of equations (i.e., functional dependencies) [18]. And in deep learning, a popular class of models called *normalizing flows* [25, 12] specify a distribution by composing a fixed distribution over some latent space, say a standard normal distribution, with a function (i.e., a functional dependence) fit to observational data. Functional dependence and independence are deeply related and interacting notions. For instance, if $B$ is a function of $A$ (written $A \twoheadrightarrow B$) and $A$ is independent of $C$ (written $A \perp\!\!\!\perp C$), then $B$ and $C$ are also independent ($B \perp\!\!\!\perp C$).[1] Moreover, dependence can be written in terms of independence: $Y$ is a function of $X$ if and only if $Y$ is conditionally independent of itself given $X$ (i.e., $X \twoheadrightarrow Y$ iff $Y \perp\!\!\!\perp Y \mid X$). Traditional graph-based languages such as Bayesian Networks (BNs) and Markov Random Fields (MRFs) cannot capture these relationships. Indeed, the graphoid axioms (which describe BNs and MRFs) [21] and axioms for conditional independence [17], do not even consider statements like $A \perp\!\!\!\perp A$ to be syntactically valid. Yet such statements

---

[1]This well-known fact (Lemma 10) is formalized and proved in Appendix A, where all proofs can be found.

38th Conference on Neural Information Processing Systems (NeurIPS 2024).

are perfectly meaningful, and reflect a deep relationship between independence, dependence, and generalizations of both notions (grounded in information theory, a point we will soon revisit).

This paper provides a simple yet expressive graphical language for describing qualitative structure such as dependence and independence in probability distributions. The idea is to specify the inputs and outputs of a set of *independent mechanisms*: processes by which some target variables $T$ are determined as a (possibly randomized) function of some source variables $S$. This idea generalizes intuition going back to Pearl [18] by allowing, for example, two mechanisms to share a target variable. So at a qualitative level, the modeler specifies not a (directed) graph, but a (directed) *hypergraph*.

If we were interested in a concrete probabilistic model, we would also need to annotate this hypergraph with quantitative information describing the mechanisms. For directed acyclic graphs, there are two standard approaches: supply conditional probability distributions (cpds) to get a BN, or supply equations to get a causal model. Correspondingly, there are two approaches to probabilistic modeling based on hypergraphs. The analogue of the first approach—supplying a probability $P(T|S)$ for each mechanism—leads to the notion of a *probabilistic dependency graph (PDG)* [23, 22, 24]. The analogue of the second approach—supplying an equation describing $T$ as a function of $S$ and independent random noise—leads to a novel generalization of a causal model (Definition 4). Models of either kind are of interest to us only insofar as they explain how hypergraphs encode qualitative aspects of probability. Qualitative information in a PDG was characterized by Richardson and Halpern [23] using a scoring function that, despite having some attractive properties, lacks justification and has not been fully understood. In particular, the PDG formalism does not appear to answer a basic question: *what does it mean for a distribution to be compatible with a directed hypergraph structure?*

We develop precisely such a notion (Definition 2) of compatibility between a distribution $\mu$ and a directed hypergraph qualitatively representing a collection of independent mechanisms—or, for short, simply (QIM-)compatibility. This definition allows us to use directed hypergraphs as a language for specifying structure in probability distributions, of which the semantics of qualitative BNs are a special case (Theorem 1). Yet QIM-compatibility can do far more than represent conditional independencies in acyclic networks. For one thing, it can encode arbitrary functional dependencies (Theorem 2); for another, it gives meaningful semantics to cyclic models. Indeed, compatibility lets us go well beyond capturing dependence and independence. The fact that Pearl [18] views causal models as representing independent mechanisms suggests that there might be a connection to causality. In fact, there is. A *witness* that a distribution $\mu$ is compatible with a hypergraph $\mathcal{A}$ is an extended distribution $\bar{\mu}$ that is nearly equivalent to (and guarantees the existence of) a causal model that explains $\mu$ with dependency structure $\mathcal{A}$ (Propositions 3 to 5). As we shall see, thinking in terms of witnesses and compatibility allows us to tie together causality, dependence, and independence.

Perhaps surprisingly, compatibility also has deep connections with information theory (Section 4). The conditional independencies of a BN can be viewed as a certain kind of information-theoretic constraint. Our notion of compatibility with a hypergraph $\mathcal{A}$ turns out to imply a generalization of this constraint (closely related to the qualitative PDG scoring function) that is meaningful for all hypergraphs (Theorem 7). Applied to cyclic models, it yields a causally inspired notion of pairwise interaction that clarifies some important misunderstandings in information theory (Examples 5 and 6).

Saying that one approach to qualitative graphical modeling has connections to so many different notions is a rather bold claim. We spend the rest of the paper justifying it.

## 2 Qualitative Independent-Mechanism (QIM) Compatibility

In this section, we present the central definition of our paper: a way of making precise Pearl's notion of "independent mechanisms", used to motivate Bayesian Networks from a causal perspective. Pearl [19, p.22] states that *"each parent-child relationship in a causal Bayesian network represents a stable and autonomous physical mechanism."* But, technically speaking, a parent-child relationship only partially describes the mechanism. Instead, the autonomous mechanism that determines the child is really represented by that child's joint relationship with all its parents. So, the qualitative aspect of a mechanism is best represented as a directed *hyperarc* [5], that can have multiple sources.

**Definition 1.** A *directed hypergraph* (or simply a hypergraph, since all our hypergraphs will be directed) consists of a set $\mathcal{N}$ of nodes and a set $\mathcal{A}$ of directed hyperedges, or *hyperarcs*; each hyperarc $a \in \mathcal{A}$ is associated with a set $S_a \subseteq \mathcal{N}$ of source nodes and a set $T_a \subseteq \mathcal{N}$ of target nodes. We write $S \xrightarrow{a} T \in \mathcal{A}$ to specify a hyperarc $a \in \mathcal{A}$ together with its sources $S = S_a$ and targets $T = T_a$. Nodes

that are neither a source nor a target of any hyperarc will seldom have any effect on our constructions; the other nodes can be recovered from the hyperarcs (by selecting $\mathcal{N} := \bigcup_{a \in \mathcal{A}} S_a \cup T_a$). Thus, we often leave $\mathcal{N}$ implicit, referring to the hypergraph simply as $\mathcal{A}$. □

Following the graphical models literature, we are interested in hypergraphs whose nodes represent variables, so that each $X \in \mathcal{N}$ will ultimately be associated with a (for simplicity, finite) set $\mathrm{V}(X)$ of possible values. However, one should not think of $\mathrm{V}$ as part of the information carried by the hypergraph. It makes perfect sense to say that $X$ and $Y$ are independent without specifying the possible values of $X$ and $Y$. Of course, when we talk concretely about a distribution $\mu$ on a set of variables $\mathcal{X} \cong (\mathcal{N}, \mathrm{V})$, those variables must have possible values—but the *qualitative* properties of $\mu$, such as independence, can be expressed purely in terms of $\mathcal{N}$, without reference to $\mathrm{V}$.

Intuitively, we expect a joint distribution $\mu(\mathcal{X})$ to be qualitatively compatible with a set of independent mechanisms (whose structure is given by a hypergraph $\mathcal{A}$) if there is a mechanistic explanation of how each target arises as a function of the variable(s) on which it depends and independent random noise. This is made precise by the following definition.

**Definition 2** (QIM-compatibility). Let $\mathcal{X}$ and $\mathcal{Y}$ be (possibly identical) sets of variables, and $\mathcal{A} = \{S_a \xrightarrow{a} T_a\}_{a \in \mathcal{A}}$ be a hypergraph with nodes $\mathcal{X}$. We say a distribution $\mu(\mathcal{Y})$ is *qualitatively independent-mechanism compatible*, or (QIM-)compatible, with $\mathcal{A}$ (symbolically: $\mu \models \Diamond \mathcal{A}$) iff there exists an extended distribution $\bar{\mu}(\mathcal{Y} \cup \mathcal{X} \cup \mathcal{U}_\mathcal{A})$ of $\mu(\mathcal{Y})$ to $\mathcal{X}$ and to $\mathcal{U}_\mathcal{A} = \{U_a\}_{a \in \mathcal{A}}$, an additional set of "noise" variables (one variable per hyperarc) according to which:

(a) the variables $\mathcal{Y}$ are distributed according to $\mu$ $\qquad$ (i.e., $\bar{\mu}(\mathcal{Y}) = \mu(\mathcal{Y})$),

(b) the variables $\mathcal{U}_\mathcal{A}$ are mutually independent $\qquad$ (i.e., $\bar{\mu}(\mathcal{U}_\mathcal{A}) = \prod_{a \in \mathcal{A}} \bar{\mu}(U_a)$ ), and

(c) the target variable(s) $T_a$ of each hyperarc $a \in \mathcal{A}$ are determined by $U_a$ and the source variable(s) $S_a$ $\qquad$ (i.e., $\forall a \in \mathcal{A}. \ \bar{\mu} \models (S_a, U_a) \twoheadrightarrow T_a$).

We call such a distribution $\bar{\mu}(\mathcal{X} \cup \mathcal{Y} \cup \mathcal{U}_\mathcal{A})$ a *witness* that $\mu$ is QIM-compatible with $\mathcal{A}$. □

While Definition 2 requires the noise variables $\{U_a\}_{a \in \mathcal{A}}$ to be independent of one another, note that they need not be independent of any variables in $\mathcal{X}$. In particular, $U_a$ may not be independent of $S_a$, and so the situation can diverge from what one would expect from a randomized algorithm, whose randomness $U$ is assumed to be independent of its input $S$. Furthermore, the variables in $\mathcal{U}$ may not be independent of one another conditional on the value of some $X \in \mathcal{X}$.

**Example 1.** $\mu(X, Y)$ is compatible with $\mathcal{A} = \{\emptyset \xrightarrow{1} \{X\}, \emptyset \xrightarrow{2} \{Y\}\}$ (depicted in PDG notation as $\rightarrow \boxed{X} \ \boxed{Y} \leftarrow$ ) iff $X$ and $Y$ are independent, i.e., $\mu(X, Y) = \mu(X)\mu(Y)$. For if $U_1$ and $U_2$ are independent and respectively determine $X$ and $Y$, then $X$ and $Y$ must also be independent. △

This is a simple illustration of a more general phenomenon: when $\mathcal{A}$ describes the structure of a Bayesian Network (BN), then QIM-compatibility with $\mathcal{A}$ coincides with satisfying the independencies of that BN (which are given, equivalently, by the *ordered Markov properties* [14], *factoring* as a product of probability tables, or *d-separation* [6]). To state the general result (Theorem 1), we must first clarify how the graphs of standard graphical and causal models give rise to directed hypergraphs.

Suppose that $G = (V, E)$ is a graph, whose edges may be directed or undirected. Given a vertex $u \in V$, write $\mathbf{Pa}_G(u) := \{v : (v, u) \in E\}$ for the set of vertices that can "influence" $u$. There is a natural way to interpret the graph $G$ as giving rise to a set of mechanisms: one for each variable $u$, which determines the value of $u$ based the values of the variables on which $u$ can depend. Formally, let $\mathcal{A}_G := \left\{ \mathbf{Pa}_G(u) \xrightarrow{u} \{u\} \right\}_{u \in V}$ be the hypergraph *corresponding* to the graph $G$.

**Theorem 1.** *If $G$ is a directed acyclic graph and $\mathcal{I}(G)$ consists of the independencies of its corresponding Bayesian network, then $\mu \models \Diamond \mathcal{A}_G$ if and only if $\mu$ satisfies $\mathcal{I}(G)$.*

Theorem 1 shows, for hypergraphs that correspond to directed acyclic graphs (dags), our definition of compatibility reduces exactly to the well-understood independencies of BNs. This means that QIM-compatibility, a notion based on the independence of causal mechanisms, gives us a very different way of characterizing these independencies—one that can be generalized to a much larger class of graphical models that includes, for example, cyclic variants [1]. Moreover, QIM-compatibility can capture properties other than independence. As the next example shows, it can capture determinism.

**Example 2.** If $\mathcal{A} = \xrightarrow{1} \boxed{X} \xleftarrow{2}$ consists of just two hyperarcs pointing to a single variable $X$, then a distribution $\mu(X)$ is QIM-compatible with $\mathcal{A}$ iff $\mu$ places all mass on a single value $x \in \mathrm{V}(X)$. △

Intuitively, if two independent coins always give the same answer (the value of $X$), then neither coin can be random. This simple example shows that we can capture determinism with multiple hyperarcs pointing to the same variable. Such hypergraphs do not correspond to graphs; recall that in a BN, two arrows pointing to $X$ (e.g., $Y \to X$ and $Z \to X$) represent a single mechanism by which $X$ is jointly determined (by $Y$ and $Z$), rather than two distinct mechanisms.

Given a hypergraph $\mathcal{A} = (\mathcal{N}, \mathcal{A})$, $X, Y \subseteq \mathcal{N}$, and a natural number $n \geq 0$, let $\mathcal{A} \sqcup_{X \to Y}^{(+n)}$ denote the hypergraph that results from augmenting $\mathcal{A}$ with $n$ additional (distinct) hyperarcs from $X$ to $Y$.

**Theorem 2.**  (a) $\mu \models X \twoheadrightarrow Y \wedge \Diamond \mathcal{A}$ *if and only if* $\forall n \geq 0.\ \mu \models \Diamond \mathcal{A} \sqcup_{X \to Y}^{(+n)}$ .

  (b) *if* $\mathcal{A} = \mathcal{A}_G$ *for a dag G, then* $\mu \models X \twoheadrightarrow Y \wedge \Diamond \mathcal{A}$ *if and only if* $\mu \models \Diamond \mathcal{A} \sqcup_{X \to Y}^{(+1)}$.

  (c) *if* $\exists a \in \mathcal{A}$ *such that* $S_a = \emptyset$ *and* $X \in T_a$, *then* $\mu \models X \twoheadrightarrow Y \wedge \Diamond \mathcal{A}$ *iff* $\mu \models \Diamond \mathcal{A} \sqcup_{X \to Y}^{(+2)}$.

Based on the intuition given after Example 2, it may seem unnecessary to ever add more than two parallel hyperarcs to ensure functional dependence in part (a). However, this intuition implicitly assumes that the randomness $U_1$ and $U_2$ of the two mechanisms is independent conditional on $X$, which may not be the case. See Appendix D for counterexamples.

Finally, as mentioned above, QIM-compatibility gives meaning to cyclic structures, a topic that we will revisit often in Sections 3 and 4. We start with a simple example.

**Example 3.** Every $\mu(X, Y)$ is compatible with $\boxed{X} \rightleftarrows \boxed{Y}$, because every distribution is compatible with $\to \boxed{X} \to \boxed{Y}$, and a mechanism with no inputs is a special case of one that can depend on $Y$. △

The logic above is an instance of an important reasoning principle, which we develop in Appendix B. Although the 2-cycle in Example 3 is straightforward, generalizing it even slightly to a 3-cycle raises a not-so-straightforward question, whose answer will turn out to have surprisingly broad implications.

**Example 4.** What $\mu(X, Y, Z)$ are compatible with the 3-cycle shown, on the right? By the reasoning above, among them must be all distributions consistent with a linear chain $\to X \to Y \to Z$. Thus, any distribution in which two variables are conditionally independent given the third is compatible with the 3-cycle. Are there distributions that are *not* compatible with this hypergraph? It is not obvious. We return to this in Section 4. △ 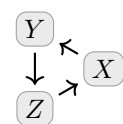

Because QIM-compatibility applies to cyclic structures, one might wonder if it also captures the independencies of undirected models. Our definition of $\mathcal{A}_G$, as is common, implicitly identifies a undirected edge $A - B$ with the pair $\{A \to B, B \to A\}$ of directed edges; in this way, it naturally converts even an *undirected* graph $G$ to a (directed) hypergraph. Compatibility with $\mathcal{A}_G$, however, does not coincide with any of the standard Markov properties corresponding to $G$ [13]. This may appear to be a flaw in Definition 2, but it is unavoidable (see Appendix B) if we wish to also capture causality, as we do in the next section.

## 3 QIM-Compatibility and Causality

Recall that in the definition of QIM-compatibility, each hyperarc represents an independent mechanism. Equations in a causal model are also viewed as representing independent mechanisms. This suggests a possible connection between the two formalisms, which we now explore. We will show that QIM-compatibility with $\mathcal{A}$ means exactly that a distribution can be generated by a causal model with the corresponding dependency structure (Section 3.1). Moreover, such causal models and QIM-compatibility witnesses are themselves closely related (Section 3.2). In this section, we establish a causal grounding for QIM-compatibility. To do so, we must first review some standard definitions.

**Definition 3** (Pearl [19]). A *structural equations model* (SEM) is a tuple $M = (\mathcal{U}, \mathcal{V}, \mathcal{F})$, where

- $\mathcal{U}$ is a set of exogenous variables;
- $\mathcal{V}$ is a set of endogenous variables (disjoint from $\mathcal{U}$);
- $\mathcal{F} = \{f_Y\}_{Y \in \mathcal{V}}$ associates to each endogenous variable $Y$ an *equation* $f_Y : \mathrm{V}(\mathcal{U} \cup \mathcal{V} - Y) \to \mathrm{V}(Y)$ that determines its value as a function of the other variables.  □

In a SEM $M$, a variable $X \in \mathcal{V}$ *does not depend* on $Y \in \mathcal{V} \cup \mathcal{U}$ if $f_X(\ldots, y, \ldots) = f_X(\ldots, y', \ldots)$ for all $y, y' \in \mathrm{V}(Y)$. Let the parents $\mathbf{Pa}_M(X)$ of $X$ be the set of variables on which $X$ depends. $M$ is *acyclic* iff $\mathbf{Pa}_M(X) \cap \mathcal{V} = \mathbf{Pa}_G(X)$ for some dag $G$ with vertices $\mathcal{V}$. In an acyclic SEM, it is

easy to see that a setting of the exogenous variables determines the values of the endogenous variables (symbolically: $M \models \mathcal{U} \twoheadrightarrow \mathcal{V}$). A *probabilistic SEM* (PSEM) $\mathcal{M} = (M, P)$ is a SEM, together with a probability $P$ over the exogenous variables. When $\mathcal{M} \models \mathcal{U} \twoheadrightarrow \mathcal{V}$, the distribution $P(\mathcal{U})$ extends uniquely to a distribution over $V(\mathcal{V} \cup \mathcal{U})$. A cylic PSEM, however, may induce more than one such distribution, or none at all. In general, a PSEM $\mathcal{M}$ induces a (possiby empty) convex set of distributions over $V(\mathcal{U} \cup \mathcal{V})$. This set is defined by two (linear) constraints: the equations $\mathcal{F}$ must hold with probability 1, and the marginal probability over $\mathcal{U}$ must equal $P$. Given a PSEM $\mathcal{M}$, let $\{\!\!\{\mathcal{M}\}\!\!\}$ consist of all joint distributions $\nu(\mathcal{U}, \mathcal{V})$ that satisfy the two constraints above; this set captures the behavior of $\mathcal{M}$ in the absence of interventions. A joint distribution $\mu(\mathbf{X})$ over $\mathbf{X} \subseteq \mathcal{V} \cup \mathcal{U}$ *can arise from* a (P)SEM $\mathcal{M}$ iff there is some $\nu \in \{\!\!\{\mathcal{M}\}\!\!\}$ whose marginal on $\mathbf{X}$ is $\mu$.

We now review the syntax of a language for describing causality. A *basic causal formula* is one of the form $[\mathbf{Y} \leftarrow \mathbf{y}]\varphi$, where $\varphi$ is a Boolean expression over the endogenous variables $\mathcal{V}$, $\mathbf{Y} \subseteq \mathcal{V}$ is a subset of them, and $\mathbf{y} \in V(\mathbf{Y})$. The language then consists of all Boolean combinations of basic formulas. In a causal model $M$ and context $\mathbf{u} \in V(\mathcal{U})$, a Boolean expression $\varphi$ over $\mathcal{V}$ is true iff it holds for all $(\mathbf{u}, \mathbf{x}) \in V(\mathcal{U}, \mathcal{V})$ consistent with the equations of $M$. Basic causal formulas are then given semantics by $(M, \mathbf{u}) \models [\mathbf{Y} \leftarrow \mathbf{y}]\varphi$ iff $(M_{\mathbf{Y} \leftarrow \mathbf{y}}, \mathbf{u}) \models \varphi$, where $M_{\mathbf{Y} \leftarrow \mathbf{y}}$ is the result of changing each $f_Y$, for $Y \in \mathbf{Y}$, to the constant function $\mathbf{s} \mapsto \mathbf{y}[Y]$, which returns (on all inputs $\mathbf{s}$) the value of $Y$ in the joint setting $\mathbf{y}$. The dual formula $\langle \mathbf{Y} \leftarrow \mathbf{y} \rangle \varphi := \neg[\mathbf{Y} \leftarrow \mathbf{y}]\neg\varphi$ is equivalent to $[\mathbf{Y} \leftarrow \mathbf{y}]\varphi$ in SEMs where each context $\mathbf{u}$ induces a unique setting of the endogenous variables [7]. A PSEM $\mathcal{M} = (M, P)$ assigns probabilities to causal formulas according to $\Pr_{\mathcal{M}}(\varphi) := P(\{\mathbf{u} \in V(\mathcal{U}) : (M, \mathbf{u}) \models \varphi\})$.

Some authors assume that for each variable $X$, there is a special "independent noise" exogenous variable $U_X$ on which only the equation $f_X$ can depend; we call a PSEM $(M, P)$ *randomized* if it contains such exogenous variables that are mutually independent according to $P$, and *fully randomized* if all its exogenous variables are of this form. Randomized PSEMs are clearly a special class of PSEMs, but note also that every PSEM can be converted to an equivalent randomized PSEM by extending it with additional dummy variables $\{U_X\}_{X \in \mathcal{V}}$ that can take only a single value. Thus, we do not lose expressive power by using randomized PSEMs. In fact, *qualitatively*, randomized PSEMs are more expressive: they can encode independence.

### 3.1 The Equivalence Between QIM-Compatibility and Randomized PSEMs

We are now equipped to formally describe the connection between QIM-compatibility and causality. At a high level, this connection should be unsurprising: witnesses and causal models both relate dependency structures to distributions, but in "opposite directions". QIM-compatibility starts with distributions and asks what dependency structures they are compatible with. Causal models, on the other hand, are explicit (quantitative) representations of dependency structures that give rise to sets of distributions. We now show that the existence of a causal model coincides with the existence of a witness. We start by showing this for the hypergraphs generated by graphs (like Bayesian networks, except possibly cyclic), which we show correspond to fully randomized causal models (Proposition 3). We then give a natural generalization of a causal model that exactly captures QIM-compatibility with an arbitrary hypergraph (Proposition 4). In both cases, the high-level result is the same: $\mu \models \mathcal{A}$ iff there is a causal model that "has dependency structure $\mathcal{A}$" that gives rise to $\mu$.

More precisely, a randomized causal model $\mathcal{M}$ *has dependency structure* $\mathcal{A}$ iff there is a 1-1 correspondence between $a \in \mathcal{A}$ and the equations of $\mathcal{M}$, such that the equation $f_a$ produces a value of $T_a$ and depends only on $S_a$ and $U_a$. The definition above emphasizes the hypergraph; for readers interested in causality, here is an equivalent one that emphasizes the causal model: $\mathcal{M}$ is of dependency structure $\mathcal{A}$ iff the targets of $\mathcal{A}$ are disjoint singletons corresponding to the elements of $\mathcal{V}$ (so $\mathcal{A} = \{S_Y \to \{Y\}\}_{Y \in \mathcal{V}}$), and $\mathbf{Pa}_{\mathcal{M}}(Y) \subseteq S_Y \cup \{U_Y\}$ for all $Y \in \mathcal{V}$. We start by presenting the result in the case where $\mathcal{A}$ corresponds to a directed graph.

**Proposition 3.** *Given a graph $G$ and a distribution $\mu$, $\mu \models \Diamond \mathcal{A}_G$ iff there exists a fully randomized PSEM of dependency structure $\mathcal{A}_G$ from which $\mu$ can arise.*

In other words, compatibility with a hypergraph corresponding to a graph means arising from a fully randomized PSEM of the appropriate dependency structure. In light of this, Theorem 1 can be viewed as formalizing a phenomenon that seems to be almost universally implicitly understood: every acyclic fully randomized SEM induces a distribution with the independencies of the corresponding Bayesian Network. Conversely, every distribution with those independencies arises from such a causal model. Both halves have been recognized before. Druzdzel and Simon [4, Theorem 1] arguably establish

one direction of the correspondence (turning a BN into a causal model), but their statement of the result obscures the possibility of a converse.[2] Pearl's *causal Markov condition* [20, Theorem 1], on the other hand, is closely related to that converse (as will be made explicit by our Proposition 3). Yet, to the best of our knowledge, the two results have not before been combined and recognized as an equivalent characterization of a BN's conditional independencies.

Like before, QIM-compatibility allows us to go much futher. It is easy to extend Proposition 3 to the dependency structures of all randomized PSEMs. But what happens if $\mathcal{A}$ contains hyperarcs with overlapping targets? Here the correspondence starts to break down for a simple reason: by definition, there is at most one equation per variable in a (P)SEM; thus, no PSEM can have dependency structure $\mathcal{A}$. Nevertheless, the correspondence between witnesses and causal models persists if we simply drop the (traditional) requirement that $\mathcal{F}$ is indexed by $\mathcal{V}$. This leads us to consider a natural generalization of a (randomized) PSEM that has an arbitrary set of equations—not just one per variable.

**Definition 4.** Let $(\mathcal{N}, \mathcal{A})$ be a hypergraph. A *generalized randomized PSEM* $\mathcal{M} = (\mathcal{X}, \mathcal{U}, \mathcal{F}, P)$ *with structure* $\mathcal{A}$ consists of sets of variables $\mathcal{X}$ and $\mathcal{U} = \{U_a\}_{a \in \mathcal{A}}$, together with a set of functions $\mathcal{F} = \{f_a : \mathrm{V}(S_a) \times \mathrm{V}(U_a) \to \mathrm{V}(T_a)\}_{a \in \mathcal{A}}$, and a probability $P_a$ over each independent noise variable $U_a$. The meanings of $\{\!\{\mathcal{M}\}\!\}$ and *can arise* are the same as for a PSEM. $\qquad\square$

**Proposition 4.** $\mu \models \Diamond \mathcal{A}$ *iff there exists a generalized randomized PSEM with structure* $\mathcal{A}$ *from which* $\mu$ *can arise.*

Generalized randomized PSEMs can capture functional dependencies, and constraints. For instance, an equality (say $X = Y$) can be encoded in a generalized randomized PSEM with a second equation for $X$. Indeed, we believe that generalized randomized PSEMs can capture a wide class of constraints, and are closely related to *causal models with constraints* [2], a discussion we defer to future work.

### 3.2 Interventions and the Correspondence Between Witnesses and Causal Models

We have seen that QIM-compatibility with $\mathcal{A}$ (i.e., the existence of a witness $\bar{\mu}$) coincides exactly with the existence of a causal model $\mathcal{M}$ from which a distribution can arise. But which witnesses correspond to which causal models? The answer to this question will be critical to extend the correspondence we have given so that it can deal with interventions. Different causal models may give rise to the same distribution, yet handle interventions differently.

There are two directions of the correspondence. Given a randomized PSEM $\mathcal{M}$, distributions arising from it are compatible with its dependency structure, and the corresponding witnesses are exactly the distributions in $\{\!\{\mathcal{M}\}\!\}$ (see Appendix E). In particular, if $\mathcal{M}$ is acyclic, there is a unique witness. The converse is more interesting: how can we turn a witness into a causal model?

**Construction 5.** Given a witness $\bar{\mu}(\mathcal{X})$ to compatibility with a hypergraph $\mathcal{A}$ with disjoint targets, construct a PSEM according to the following (non-deterministic) procedure. Take $\mathcal{V} := \cup_{a \in \mathcal{A}} T_a$, $\mathcal{U} := \mathcal{U}_\mathcal{A} \cup (\mathcal{X} - \mathcal{V})$, and $P(\mathcal{U}) := \bar{\mu}(\mathcal{U})$. For each $X \in \mathcal{V}$, there is a unique $a_X \in \mathcal{A}$ whose targets $T_{a_X}$ contain $X$. Since $\bar{\mu} \models (U_{a_X}, S_{a_X}) \twoheadrightarrow T_{a_X}$ (this is just property (c) in Definition 2), $X \in T_{a_X}$ must also be a function of $S_{a_X}$ and $U_{a_X}$; take $f_X$ to be such a function. More precisely, for each $u \in \mathrm{V}(U_{a_X})$ and $\mathbf{s} \in \mathrm{V}(S_{a_X})$ for which $\bar{\mu}(U_{a_X} = u, S_{a_X} = \mathbf{s}) > 0$, there is a unique $t \in \mathrm{V}(T_{a_X})$ such that $\bar{\mu}(u, \mathbf{s}, t) > 0$. In this case, set $f_X(u, \mathbf{s}, \ldots) := t[X]$. If $\bar{\mu}(U_{a_X} = u, S_{a_X} = \mathbf{s}) = 0$, $f_X(u, \mathbf{s}, \ldots)$ can be an arbitrary function of $u$ and $\mathbf{s}$. Let $\mathrm{PSEMs}_\mathcal{A}(\bar{\mu})$ denote the set of PSEMs that can result.

It's clear from Construction 5 that $\mathrm{PSEMs}_\mathcal{A}(\bar{\mu})$ is always nonempty, and is a singleton iff $\bar{\mu}(u, s) > 0$ for all $(a, u, s) \in \sqcup_{a \in \mathcal{A}} \mathrm{V}(U_a, S_a)$. A witness with this property exists when $\mu$ is positive (i.e., $\mu(\mathcal{X} = \mathbf{x}) > 0$ for all $\mathbf{x} \in \mathrm{V}(\mathcal{X})$), in which case the construction gives a unique causal model. Conversely, we have seen that an acylic model $\mathcal{M}$ gives rise to a unique witness. So, in the simplest cases, models $\mathcal{M}$ with structure $\mathcal{A}$ and witnesses $\bar{\mu}$ to compatibility with $\mathcal{A}$ are equivalent. But there are two important caveats.

1. A causal model $\mathcal{M}$ can contain more information than a witness $\bar{\mu}$ if some events have probability zero. For instance, $\bar{\mu}$ could be a point mass on a single joint outcome $\omega$ of all variables that satisfies the equations of $\mathcal{M}$. But $\mathcal{M}$ cannot be reconstructed uniquely from $\bar{\mu}$ because there may be many causal models for which $\omega$ is a solution.

---

[2]Indeed, Druzdzel and Simon [4] state that "a causal structure does not necessarily imply independences", suggesting that they did not realize that their result could be used to characterize BN independencies.

2. A witness $\bar{\mu}$ can contain more information than a causal model $\mathcal{M}$ if $\mathcal{M}$ is cyclic. For example, suppose that $\mathcal{M}$ consists of two variables, $X$ and $X'$, and equations $f_X(X') = X'$ and $f_{X'}(X) = X$. In this case, $\bar{\mu}$ cannot be reconstructed from $\mathcal{M}$, because $\mathcal{M}$ does not contain information about the distribution of $X$.

These two caveats appear to be very different, but they fit together in a surprisingly elegant way.

**Proposition 5.** *If $\bar{\mu}(\mathcal{X}, \mathcal{U}_{\mathcal{A}})$ is a witness for QIM-compatibility with $\mathcal{A}$ and $\mathcal{M}$ is a PSEM with dependency structure $\mathcal{A}$, then $\bar{\mu} \in \{\!\{\mathcal{M}\}\!\}$ if and only if $\mathcal{M} \in \mathrm{PSEMs}_{\mathcal{A}}(\bar{\mu})$.*

Equivalently, this means that $\mathrm{PSEMs}_{\mathcal{A}}(\bar{\mu})$, the possible outputs of Construction 5, are precisely the randomized PSEMs of dependency structure $\mathcal{A}$ that can give rise to $\bar{\mu}$. This is already substantial evidence that causal models $\mathcal{M} \in \mathrm{PSEMs}_{\mathcal{A}}(\bar{\mu})$ are closely related to the QIM-compatibility witness $\bar{\mu}$. But everything we have seen so far describes only the correspondence in the absence of intervention, a setting in which many causal models are indistinguishable. Yet the correspondence goes deeper; it extends to interventions. This claim may seem dubious, as the obvious distinction between observing and doing is a fundemental principle of causality. What could a witness, which is purely probabilistic, have to say about intervention? In a randomized PSEM $\mathcal{M}$, we can define an event

$$\mathrm{do}_{\mathcal{M}}(\mathbf{X}=\mathbf{x}) := \bigcap_{X \in \mathbf{X}} \bigcap_{\mathbf{s} \in \mathrm{V}(\mathbf{Pa}(X))} \big(f_X(U_X, \mathbf{s}) = \mathbf{x}[X]\big), \qquad \text{where } \mathbf{x}[X] \text{ is the} \atop \text{value of } X \text{ in } \mathbf{x}. \tag{1}$$

This is intuitively the event in which the randomness is such that $\mathbf{X} = \mathbf{x}$ regardless of the values of the parent variables. As we now show, conditioning $\bar{\mu}$ on $\mathrm{do}_{\mathcal{M}}(\mathbf{X}=\mathbf{x})$ has the effect of intervention—at least as long as the noise variables $\mathcal{U}_{\mathcal{A}} = \{\mathcal{U}_a\}_{a \in \mathcal{A}}$ are independent of the other exogenous variables $\mathcal{U} \setminus \mathcal{U}_{\mathcal{A}}$ in addition to one another (e.g., when $\mathcal{M}$ is fully randomized).

**Theorem 6.** *Suppose that $\bar{\mu}$ is a QIM witness for $\mathcal{A}$, that $\mathcal{M} = (\mathcal{U}, \mathcal{V}, \mathcal{F}, P) \in \mathrm{PSEMs}_{\mathcal{A}}(\bar{\mu})$ is a corresponding PSEM, and that the noise variables $\mathcal{U}_{\mathcal{A}} = \{U_X\}_{X \in \mathcal{V}}$ are independent of the other exogenous variables $\mathcal{U} \setminus \mathcal{U}_{\mathcal{A}}$. For all $\mathbf{X} \subseteq \mathcal{V}$ and $\mathbf{x} \in \mathrm{V}(\mathbf{X})$, if $\bar{\mu}(\mathrm{do}_{\mathcal{M}}(\mathbf{X}=\mathbf{x})) > 0$, then*

*(a) $\bar{\mu}(\mathcal{V} \mid \mathrm{do}_{\mathcal{M}}(\mathbf{X}=\mathbf{x}))$ can arise from $\mathcal{M}_{\mathbf{X} \leftarrow \mathbf{x}}$;*

*(b) for all events $\varphi \subseteq \mathrm{V}(\mathcal{V})$, $\mathrm{Pr}_{\mathcal{M}}\big([\mathbf{X} \leftarrow \mathbf{x}]\varphi\big) \leq \bar{\mu}\big(\varphi \mid \mathrm{do}_{\mathcal{M}}(\mathbf{X}=\mathbf{x})\big) \leq \mathrm{Pr}_{\mathcal{M}}\big(\langle \mathbf{X} \leftarrow \mathbf{x}\rangle\varphi\big)$ and all three are equal when $\mathcal{M} \models \mathcal{U} \twoheadrightarrow \mathcal{V}$ (such as when $\mathcal{M}$ is acyclic).*

Theorem 6 shows that the relationship between witnesses and causal models extends to interventions. Even when $\mathrm{do}_{\mathcal{M}}(\mathbf{X}=\mathbf{x})$ has probability zero, it is always possible to find a nearly equivalent setting where the bounds of the theorem apply.[3] Intervention and conditioning are conceptually very different, so it may seem surprising that conditioning can have the effect of intervention (and also that the Pearl's $\mathrm{do}(\cdot)$ notation actually corresponds to an event [9]). We emphasize that the conditioning (on $\mathrm{do}_{\mathcal{M}}(\mathbf{X}=\mathbf{x})$) is on the randomness $\{U_X\}_{X \in \mathbf{X}}$ and not $\mathbf{X}$ itself; intervening on $\mathbf{X}=\mathbf{x}$ is indeed fundamentally different from conditioning on $\mathbf{X}=\mathbf{x}$.

## 4 QIM-Compatibility and Information Theory

The fact that the dependency structure of a (causal) Bayesian network describes the independencies of the distribution it induces is fundamental to both causality and probability. It makes explicit the distributional consequences of BN structure. Yet, despite substantial interest [1], generalizing the BN case to more complex (e.g., cyclic) dependency structures remains largely an open problem. In Section 4.1, we generalize the BN case by providing an information-theoretic constraint, capable of capturing conditional independence, functional dependence, and more, on the distributions that can arise from an *arbitrary* dependency structure. This connection between causality and information theory has implications for both fields. It grounds the cyclic dependency structures found in causality in concrete constraints on the distributions they represent. At the same time, it allows us to resolve longstanding confusion about structure in information theory, clarifying the meaning of the so-called "interaction information", and recasting a standard counterexample to substantiate the claim it was intended to oppose. In Section 4.2, we strengthen this connection. Using entropy to measure distance to (in)dependence, we develop a scoring function to measure how far a distribution is from being QIM-compatible with a given dependency structure. This function turns out to have an intimate relationship

---

[3]More precisely, for all $\epsilon > 0$, there exists some $\mathcal{M}'$ that differs from $\mathcal{M}$ on the probabilities of all causal formulas by at most $\epsilon$, and a distribution $\bar{\mu}'$ that is $\epsilon$-close to $\bar{\mu}$, such that $\bar{\mu}'(\mathrm{do}_{\mathcal{M}'}(\mathbf{X}=\mathbf{x})) > 0$.

with the qualitative PDG scoring fucntion $IDef$, which we use to show that our information-theoretic constraints degrade gracefully on "near-compatible" distributions.

We now review the critical information theoretic concepts and their relationships to (in)dependence (see Appendix C.1 for a full primer). Conditional entropy $H_\mu(Y|X)$ measures how far $\mu$ is from satisfying the functional dependency $X \twoheadrightarrow Y$. Conditional mutual information $I_\mu(Y;Z|X)$ measures how far $\mu$ is from satisfying the conditional independence $Y \perp\!\!\!\perp Z \mid X$. Linear combinations of these quantities (for $X, Y, Z \subseteq \mathcal{X}$) can be viewed as the inner product between a coefficient vector $\mathbf{v}$ and a $2^{|\mathcal{X}|} - 1$ dimensional vector $\mathbf{I}_\mu$ that we will call the *information profile* of $\mu$. For three variables, the components of this vector are illustrated in Figure 1 (right). It is not hard to see that an arbitrary conjunction of (conditional) (in)dependencies can be expressed as a constraint $\mathbf{I}_\mu \cdot \mathbf{v} \geq 0$, for some appropriate choice of vector $\mathbf{v}$.

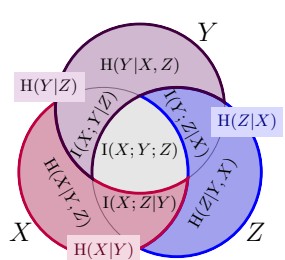

Figure 1: $\mathbf{I}_\mu$.

We now formally introduce the qualitative PDG scoring function $IDef$, which interprets a hypergraph structure $\mathcal{A}$ as a function of the form $\mathbf{I}_\mu \cdot \mathbf{v}_{\mathcal{A}}$. This *information deficiency*, given by

$$IDef_{\mathcal{A}}(\mu) = \mathbf{I}_\mu \cdot \mathbf{v}_{\mathcal{A}} := -H_\mu(\mathcal{X}) + \sum_{a \in \mathcal{A}} H_\mu(T_a \mid S_a), \tag{2}$$

is the difference between the number of bits needed to (independently) specify the randomness in $\mu$ along the hyperarcs of $\mathcal{A}$, and the number of bits needed to specify a sample of $\mu$ according to its own structure ($\emptyset \to \mathcal{X}$). While $IDef$ has some nice properties[4], it can also behave unintuitively in some cases; for instance, it can be negative. Clearly, it does not measure how close $\mu$ is to being structurally compatible with $\mathcal{A}$, in general. Nevertheless, there is still a fundamental relationship between $IDef$ and QIM-compatibility, as we now show.

## 4.1 A Necessary Condition for QIM-Compatibility

What constraints does QIM-compatibility with $\mathcal{A}$ place on a distribution $\mu$? When $G$ is a dag, we have seen that if $\mu \models \Diamond \mathcal{A}_G$, then $\mu$ must satisfy the independencies of the corresponding Bayesian network (Theorem 1); we have also seen that additional hyperarcs impose functional dependencies (Theorem 2). But these results apply only when $\mathcal{A}$ is of a very special form. More generally, $\mu \models \Diamond \mathcal{A}$ implies that $\mu$ can arise from some randomized causal model whose equations have dependency structure $\mathcal{A}$ (Propositions 3 and 4). Still, unless $\mathcal{A}$ has a particularly special form, it is not obvious whether or not this says something about $\mu$. The primary result of this section is an information-theoretic bound (Theorem 7) that generalizes most of the concrete consequences of QIM-compatibility we have seen so far (Theorems 1 and 2). The result is a connection between information theory and causality; it yields an information-theoretic test for complex causal dependency structures, and enables causal notions of structure to dispel misconceptions in information theory.

**Theorem 7.** *If $\mu \models \Diamond \mathcal{A}$, then $IDef_{\mathcal{A}}(\mu) \leq 0$.*

Theorem 7 applies to all hypergraphs, and subsumes every general-purpose technique we know of for proving that $\mu \not\models \Diamond \mathcal{A}$. Indeed, the negative directions of Theorems 1 and 2 are immediate consequences of it. To illustrate some of its subtler implications, we return to the 3-cycle in Example 4.

**Example 5.** It is easy to see (e.g., by inspecting Figure 1) that $IDef_{\text{3-cycle}}(\mu) = H_\mu(Y|X) + H_\mu(Z|Y) + H_\mu(X|Z) - H_\mu(XYZ) = -I_\mu(X;Y;Z)$. Theorem 7 therefore tells us that a distribution $\mu$ that is QIM-compatible with the 3-cycle cannot have negative interaction information $I_\mu(X;Y;Z)$. What does this mean? When $I(X;Y;Z) < 0$, conditioning on one variable causes the other two to share more information than they did before. The most extreme instance is $\mu_{xor}$, the distribution in which two variables are independent and the third is their parity (illustrated on the right). It seems intuitively clear that $\mu_{xor}$ cannot arise from the 3-cycle, a causal model with only pairwise dependencies. This is difficult to prove directly, but is an immediate consequence of Theorem 7. $\triangle$

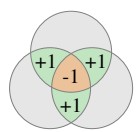

For many, there is an intuition that $I(X;Y;Z) < 0$ should require a fundamentally "3-way" interaction between the variables, and should not arise through pairwise interactions alone [10]. This has

---

[4]It captures BN independencies and the dependencies of Theorem 2, reduces to maximum entropy for the empty hypergraph, and combines with the quantitative PDG scoring function [23] to capture factor graphs.

been a source of conflict [26, 16, 15, 3], because traditional ways of making precise "pairwise inter-actions" (e.g., maximum entropy subject to pairwise marginal constraints and pairwise factorization) do not ensure that $I(X; Y; Z) \geq 0$. But QIM-compatibility does. One can verify by enumeration that the 3-cycle is the most expressive causal structure with no joint dependencies, and we have already proven that QIM-compatibility with that hypergraph implies non-negative interaction information. QIM-compatibility has another even more noteworthy clarifying effect on information theory.

There is a school of thought that contends that *all* structural information in $\mu(\mathcal{X})$ is captured by its information profile $\mathbf{I}_\mu$. This position has fallen out of favor in some communities due to standard counterexamples: distributions that have intuitively different structures yet share an information profile [11]. However, with "structure" explicated by compatibility, the prototypical counterexample of this kind suddenly supports the very notion it was meant to challenge, suggesting in an unexpected way that the information profile may yet capture the essence of probabilistic structure.

**Example 6.** Let $A$, $B$, and $C$ be variables with $\mathrm{V}(A), \mathrm{V}(B), \mathrm{V}(C) = \{0, 1\}^2$. Using independent fair coin flips $X_1$, $X_2$, and $X_3$, define two joint distributions, $P$ and $Q$, over $A, B, C$ as follows. Define $P$ by selecting $A := (X_1, X_2)$, $B := (X_2, X_3)$, and $C := (X_3, X_1)$. Define $Q$ by selecting $A := (X_1, X_2)$, $B := (X_1, X_3)$, and 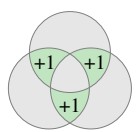 $C := (X_1, X_2 \oplus X_3)$. Structurally, $P$ and $Q$ appear to be very different. According to $P$, the first components of the three variables $(A, B, C)$ are independent, yet they are identical according to $Q$. Moreover, $P$ has only simple pairwise interactions between the variables, while $Q$ has $\mu_{xor}$ (a clear 3-way interaction) embedded within it. Yet $P$ and $Q$ have identical information profiles (see right): in both cases, each of $\{A, B, C\}$ is determined by the values of the other two, each pair share one bit of information given the third, and $I(A; B; C) = 0$.

This example has been used to argue that multivariate Shannon information does not take into account important structural differences between distributions [11]. We are now in a position to give a novel and particularly persuasive response, by appealing to QIM-compatibility. Unsurprisingly, $P$ is compatible with the 3-cycle; it is clearly consists of "2-way" interactions, as each pair of variables shares a bit. But, counterintuitively, the distribution $Q$ is *also* compatible with the 3-cycle! (The reader is encouraged to verify that $U_1 = X_3 \oplus X_1$, $U_2 = X_2$, and $U_3 = X_3$ serves as a witness.) To emphasize: this is despite the fact that $Q$ is just $\mu_{xor}$ (which is certainly not compatible with the 3-cycle) together with a seemingly irrelevant random bit $X_1$. By the results of Section 3, this means there is a causal model without joint dependence giving rise to $Q$—so, despite appearances, $Q$ does not require a 3-way interaction. Indeed, $P$ and $Q$ are QIM-compatible with precisely the same hypergraphs over $\{A, B, C\}$, suggesting that they don't have a structural difference after all. $\triangle$

In light of Example 6, one might reasonably conjecture that the converse of Theorem 7 holds. Unfortunately, it does not (see Appendix C.3); the quantity $IDef_{\mathcal{A}}(\mu)$ does not completely determine whether or not $\mu \models \Diamond\mathcal{A}$. We now pursue a new (entropy-based) scoring function that does. This will allow us to generalize Theorem 7 to distributions that are only "near-compatible" with $\mathcal{A}$.

### 4.2 A Scoring Function for QIM-Compatibility

Here is a function that measures how far a distribution $\mu$ is from being QIM-compatible with $\mathcal{A}$.

$$\mathrm{QIM}Inc_{\mathcal{A}}(\mu) := \inf_{\substack{\nu(\mathcal{U}, \mathcal{X}) \\ \nu(\mathcal{X}) = \mu(\mathcal{X})}} - \mathrm{H}_\nu(\mathcal{U}) + \sum_{a \in \mathcal{A}} \mathrm{H}_\nu(U_a) + \sum_{a \in \mathcal{A}} \mathrm{H}_\nu(T_a | S_a, U_a). \tag{3}$$

$\mathrm{QIM}Inc$ is a direct translation of Definition 2 (a-c); it measures the (optimal) quality of an extended distribution $\nu$ as a witness. The infimum restricts the search to $\nu$ satisfying (a), the first two terms measure $\nu$'s discrepancy of with (b), and the last term measures $\nu$'s discrepancy with (c). Therefore:

**Proposition 8.** $\mathrm{QIM}Inc_{\mathcal{A}}(\mu) \geq 0$, with equality iff $\mu \models \Diamond\mathcal{A}$.

Although they seem to be very different, $\mathrm{QIM}Inc$ and $IDef$ turn out to be closely related. In fact, modulo the infimum, $\mathrm{QIM}Inc_{\mathcal{A}}$ is a special case of $IDef$—not for the hypergraph $\mathcal{A}$, but rather for a transformed one $\mathcal{A}^\dagger$ that models the noise variables explcitly. To construct $\mathcal{A}^\dagger$ from $\mathcal{A}$, add new nodes $\mathcal{U} = \{U_a\}_{a \in \mathcal{A}}$, and replace each hyperarc

$$\boxed{S_a} \xrightarrow{a} \boxed{T_a} \quad \text{with the pair of hyperarcs} \quad \begin{array}{c} \xrightarrow{a_0} \boxed{U_a} \\ \boxed{S_a} \xrightarrow{a_1} \boxed{T_a} \end{array}.$$

Finally, add one additional hyperarc $\mathcal{U} \to \mathcal{X}$. (Intuitively, this hyperarc creates functional dependencies in the spirit of Theorem 2.) With these definitions in place, we can state a theorem that bounds QIM*Inc* above and below with information deficiencies (Theorem 9). The lower bound generalizes Theorem 7 by giving an upper limit on $IDef_{\mathcal{A}}(\mu)$ even for distributions $\mu$ that are not QIM-compatible with $\mathcal{A}$. The upper bound is tight in general, and shows that QIM*Inc*$_{\mathcal{A}}$ can be equivalently defined as a minimization over $IDef_{\mathcal{A}^{\dagger}}$.

**Theorem 9.** (a) *If $(\mathcal{X}, \mathcal{A})$ is a hypergraph, $\mu(\mathcal{X})$ is a distribution, and $\nu(\mathcal{X}, \mathcal{U})$ is an extension of $\nu$ to additional variables $\mathcal{U} = \{U_a\}_{a \in \mathcal{A}}$ indexed by $\mathcal{A}$, then:*

$$IDef_{\mathcal{A}}(\mu) \leq \mathrm{QIM}Inc_{\mathcal{A}}(\mu) \leq IDef_{\mathcal{A}^{\dagger}}(\nu).$$

(b) *For all $\mu$ and $\mathcal{A}$, there is a choice of $\nu$ that achieves the upper bound. That is,*

$$\mathrm{QIM}Inc_{\mathcal{A}}(\mu) = \min \left\{ \; IDef_{\mathcal{A}^{\dagger}}(\nu) : \; \begin{matrix} \nu \in \Delta \mathrm{V}(\mathcal{X}, \mathcal{U}) \\ \nu(\mathcal{X}) = \mu(\mathcal{X}) \end{matrix} \; \right\}.$$

The semantics of PDGs are based on the idea of measuring (and resolving) *inconsistency*, which is defined as a minimization over $IDef$ (plus a term that captures relevant concrete probabilistic information). Thus, Theorem 9 (b) tells us that QIM-compatibility (with $\mathcal{A}$) can be captured with a qualitative PDG (namely, $\mathcal{A}^{\dagger}$). It follows that our notion of QIM-compatibility can be viewed as a special case of the semantics of PDGs—one that, as we have shown, has a causal interpretation.

## 5   Discussion and Conclusions

We have shown how directed hypergraphs can be used to represent structural aspects of distributions. Moreover, they can do so in a way that generalizes conditional independencies and functional dependencies and has deep connections to causality and information theory. This notion of QIM-compatibility can be captured with PDGs, and also partially explains the qualitative foundations of these models. Still, many questions remain open.

Perhaps the most important open problem is that of computing whether or not a given distribution $\mu$ is QIM compatible with a directed hypergraph $\mathcal{A}$. We have implemented a rudimentary approach (based on solving problem (3) to calculate QIM*Inc*) that works in practice for small examples, but that approach scales poorly, and its correctness has not yet been proved. Even representing a distribution $\mu$ over $n$ variables requires $\Omega(2^n)$ space in general, and a candidate witness $\bar{\mu}$ is even bigger: if all variables are binary, $|\mathcal{A}| = m$, and $|S_a|, |T_a| \leq k$ for all $a \in \mathcal{A}$, then a direct implementation of (3) is a non-convex optimization problem with at most $2^{n+mk(2^k)}$ variables. Even accepting the (substantial) cost of representing extended distributions, we do not have a bound on the time needed to solve the optimization problem. There are more compact ways of representing the joint distributions $\mu$ used in practice (by assuming (in)dependencies), but we do not know if such independence assumptions make it easier to determine whether $\mu \models \Diamond \mathcal{A}$ for arbitrary $\mathcal{A}$. But computing $IDef_{\mathcal{A}}(\mu)$ can be much easier.[5] We suspect that Theorem 7, a nontrivial condition for QIM-compatibility that requires only computing $IDef_{\mathcal{A}}(\mu)$, could play a critical role in designing such an inference procedure.

Another major open problem is that of more precisely understanding the implications of QIM-compatibility in cyclic models. We do not yet know, for example, whether the same set of distributions are QIM-compatible with the clockwise and counter-clockwise 3-cycles.

As mentioned in Section 3, our notion of QIM-compatibility has led us to a generalization of a standard causal model (Definition 4). A proper investigation of this novel modeling tool (which we have not attempted in this paper) would include concrete motivating examples, a careful account of interventions and counterfactuals in this general setting, and results situating these causal models among other generalizations of causal models in the literature.

We hope to address these questions in future work.

---

[5]The complexity of calculating $IDef_{\mathcal{A}}(\mu)$ is typically dominated by the difficulty of calculating the joint entropy $\mathrm{H}(\mu)$. It can be difficult to compute $\mathrm{H}(\mu)$ in some cases (e.g., for undirected models), but in others (e.g., for Bayesian Networks or clique trees) the same assumptions that enable a compact representation of $\mu$ also make it easy to calculate $\mathrm{H}(\mu)$.

## Acknowledgments and Disclosure of Funding

We would like to thank the reviewers for useful discussion and helpful feedback, such as the pointer to Druzdzel and Simon [4], and for asking us to expand on the complexity of inference. Thank you to Matt MacDermott for identifying a bug in a prior version of Theorem 6, and to Matthias Georg Mayer for catching several low-level issues with the presentation. The work of Halpern and Richardson was supported in part by AFOSR grant FA23862114029, MURI grant W911NF-19-1-0217, ARO grant W911NF-22-1-0061, and NSF grant FMitF-2319186. S.P. is supported in part by the NSF under Grants Nos. CCF-2122230 and CCF-2312296, a Packard Foundation Fellowship, and a generous gift from Google.

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

# A  Proofs

We begin with a de-randomization construction, that will be useful for the proofs.

## A.1  From CPDs to Distributions over Functions

Compare two objects:

- a cpd $p(Y|X)$, and
- a distribution $q(Y^X)$ over functions $g : \mathrm{V}X \to \mathrm{V}Y$.

The latter is significantly larger — if both $|\mathrm{V}X| = |\mathrm{V}Y| = N$, then $q$ is a $N^N$ dimensional object, while $p$ is only dimension $N^2$. A choice of distribution $q(Y^X)$ corresponds to a unique choice cpd $p(Y|X)$, according to

$$p(Y{=}y \mid X{=}x) := q(Y^X(x) = y).$$

**Claim 1.**    *1. The definition above in fact yields a cpd, i.e., $\sum_y p(Y{=}y|X{=}x) = 1$ for all*
*$x \in \mathrm{V}X$.*
*2. This definition of $p(Y|X)$ is the conditional marginal of any joint distribution $\mu(X, Y, Y^X)$*
*satisfying $\mu(Y^X) = q$ and $\mu(Y = Y^X(X)) = 1$.*

Both $p$ and $q$ give probabilistic information about $Y$ conditioned on $X$. But $q(Y^X)$ contains strictly more information. Not only does it specify the distribution over $Y$ given $X{=}x$, but it also contains counter-factual information about the distribution of $Y$ if $X$ were equal to $x'$, conditioned on the fact that, in reality, $X{=}x$.

Is there a natural construction that goes in the opposite direction, intuitively making as many independence assumptions as possible? It turns out there is:

$$q(Y^X{=}g) = \prod_{x \in \mathrm{V}X} p(Y{=}g(x) \mid X{=}x).$$

Think of $Y^X$ as a collection of variables $\{Y^x : x \in \mathrm{V}X\}$ describing the value of the function for each input, so that $q$ is a joint distribution over them. This construction simply asks that these variables be independent. Specifying a distribution with these independences amounts to a choice of "marginal" distribution $q(Y^x)$ for each $x \in \mathrm{V}X$, and hence is essentially a funciton of type $\mathrm{V}X \to \Delta \mathrm{V}Y$, the same as $p$. In addition, if we apply the previous construction, we recover $p$, since:

$$
\begin{aligned}
q(Y^X(x) = y) &= \sum_{g:\mathrm{V}X\to\mathrm{V}Y} \mathbb{1}[g(x) = y] \prod_{x'\in\mathrm{V}X} p(Y{=}g(x') \mid X{=}x')\\
&= \sum_{g:\mathrm{V}X\to\mathrm{V}Y} \mathbb{1}[g(x) = y] p(Y{=}g(x) \mid X{=}x) \prod_{x'\neq x} p(Y{=}g(x') \mid X{=}x')\\
&= p(Y{=}y \mid X{=}x) \sum_{g:\mathrm{V}X\to\mathrm{V}Y} \mathbb{1}[g(x) = y] \prod_{x'\neq x} p(Y{=}g(x') \mid X{=}x')\\
&= p(Y{=}y \mid X{=}x) \sum_{g:\mathrm{V}X\setminus\{x\}\to\mathrm{V}Y} \prod_{x'\in\mathrm{V}X\setminus\{x\}} p(Y{=}g(x') \mid X{=}x')\\
&= p(Y{=}y \mid X{=}x).
\end{aligned}
$$

The final equality holds because the remainder of the terms can be viewed as the probability of selecting any function from $X \setminus \{x\}$ to $Y$, under an analogous measure; thus, it equals 1. This will be a useful construction for us in general.

## A.2  Results on (In)dependence

**Lemma 10.** *Suppose $X_1, \ldots, X_n$ are variables, $Y_1, \ldots, Y_n$ are sets, and for each $i \in \{1, \ldots n\}$, we have a function $f_i : \mathrm{V}(X_i) \to Y_i$. Then if $X_1, \ldots, X_n$ are mutually independent (according to a joint distribution $\mu$), then so are $f_1(X_1), \ldots, f_n(X_n)$.*

*Proof.* This is an intuitive fact, but we provide a proof for completeness. Explicitly, mutual independence of $X_1, \ldots, X_n$ means that, for all joint settings $\mathbf{x} = (x_1, \ldots x_n)$, we have $\mu(X_1=x_1, \ldots, X_n=x_n) = \prod_{i=1}^{n} \mu(X_i=x_i)$. So, for any joint setting $\mathbf{y} = (y_1, \ldots, y_n) \in Y_1 \times \cdots \times Y_n$, we have

$$
\mu\Big(f_1(X_1)=y_1, \ldots, f_n(X_n)=y_n\Big) = \mu(\{\mathbf{x} : \mathbf{f}(\mathbf{x}) = \mathbf{y}\})
$$

$$
= \sum_{\substack{(x_1,\ldots,x_n)\in V(X_1,\ldots,X_n) \\ f_1(x_1)=y_1, \,\ldots,\, f_n(x_n)=y_n}} \mu(X_1=x_1, \ldots, X_n=x_n)
$$

$$
= \sum_{\substack{x_1 \in VX_1 \\ f_1(x_1)=y_1}} \cdots \sum_{\substack{x_n \in VX_n \\ f_n(x_n)=y_n}} \mu(X_1=x_1, \ldots, X_n=x_n)
$$

$$
= \sum_{\substack{x_1 \in VX_1 \\ f_1(x_1)=y_1}} \cdots \sum_{\substack{x_n \in VX_n \\ f_n(x_n)=y_n}} \prod_{i=1}^{n} \mu(X_i=x_i)
$$

$$
= \left( \sum_{\substack{x_1 \in VX_1 \\ f_1(x_1)=y_1}} \mu(X_1=x_1) \right) \cdots \left( \sum_{\substack{x_n \in VX_n \\ f_n(x_n)=y_n}} \mu(Y_1 = y_1) \right)
$$

$$
= \prod_{i=1}^{n} \mu(f_i(X_i) = y_i). \qquad \square
$$

**Lemma 11** (properties of determination)**.**

 1. *If $\nu \models A \twoheadrightarrow B$ and $\nu \models A \twoheadrightarrow C$, then $\nu \models A \twoheadrightarrow (B, C)$.*

 2. *If $\nu \models A \twoheadrightarrow B$ and $\nu \models B \twoheadrightarrow C$, then $\nu \models A \twoheadrightarrow C$.*

*Proof.* $\nu \models X \twoheadrightarrow Y$, means there exists a function $f : V(A) \to V(B)$ such that $\nu(f(Y) = X) = 1$, i.e., the event $f(A) = B$ occurs with probability 1.

 1. Let $f : V(A) \to V(B)$ and $g : V(A) \to V(C)$ be such that $\nu(f(A) = B) = 1 = \nu(g(A) = C)$. Since both events happen with probability 1, so must the event $f(A) = B \cap g(A) = C$. Thus the event $(f(A), g(A)) = (B, C)$ occurs with probability 1. Therefore, $\nu \models A \twoheadrightarrow (B, C)$.

 2. The same ideas, but faster: we have $f : V(A) \to V(B)$ as before, and $g : V(B) \to V(C)$, such that the events $f(A) = B$ and $g(B) = C$ occur with proability 1. By the same logic, it follows that their conjunction holds with probability 1, and hence $C = f(g(A))$ occurs with probability 1. So $\nu \models A \twoheadrightarrow C$.

$$\square$$

**Theorem 1.** *If $G$ is a directed acyclic graph and $\mathcal{I}(G)$ consists of the independencies of its corresponding Bayesian network, then $\mu \models \Diamond \mathcal{A}_G$ if and only if $\mu$ satisfies $\mathcal{I}(G)$.*

*Proof.* Label the vertics of $G = (\mathcal{N}, E)$ by natural numbers so that they are a topological sort of $G$—that is, without loss of generality, suppose $\mathcal{N} = [n] := \{1, 2, \ldots, n\}$, and $i < j$ whenever $i \to j \in E$. By the definition of $\mathcal{A}_G$, the arcs $\mathcal{A}_G = \{S_i \xrightarrow{i} i\}_{i=1}^{n}$ are also indexed by integers. Finally, write $\mathcal{X} = (X_1, \ldots, X_n)$ for the variables $\mathcal{X}$ corresponding to $\mathcal{N}$ over which $\mu$ is defined.

( $\implies$ ). Suppose $\mu \models \Diamond \mathcal{A}_G$. This means there is an extension of $\bar{\mu}(\mathcal{X}, \mathcal{U})$ of $\mu(\mathcal{X})$ to additional independent variables $\mathcal{U} = (U_1, \ldots, U_n)$, such that $\bar{\mu} \models (S_i, U_i) \twoheadrightarrow i$ for all $i \in [n]$.

First, we claim that if $\bar{\mu}$ is such a witness, then $\bar{\mu} \models (U_1, \ldots, U_k) \twoheadrightarrow (X_1, \ldots, X_k)$ for all $k \in [n]$, and so in particular, $\bar{\mu} \models \mathcal{U} \twoheadrightarrow \mathcal{X}$. This follows from QIM-compatibility's condition (c) and the fact that $G$ is acyclic, by induction. In more detail: The base case of $k = 0$ holds vacuously. Suppose that

$\bar{\mu} \models (X_1, \ldots, X_k)$ for some $k < n$. Now, conditon (c) of Definition 2 says $\bar{\mu} \models (S_{k+1}, U_{k+1}) \twoheadrightarrow X_{k+1}$. Because the varaibles are sorted in topological order, the parent variables $S_{k+1}$ are a subset of $\{X_1, \ldots, X_n\}$, which are determined by $\mathcal{U}$ by the induction hypothesis; at the same time clearly $\bar{\mu} \models (U_1, \ldots, U_{k+1}) \twoheadrightarrow U_{k+1}$ as well. So, by two instances of Lemma 11, $\bar{\mu} \models (U_1, \ldots U_{k+1}) \twoheadrightarrow X_{k+1}$. Combining with our inductive hypothesis, we find that $\bar{\mu} \models (U_1, \ldots U_{k+1}) \twoheadrightarrow (X_1, \ldots, X_{k+1})$. So, by induction, $\bar{\mu} \models (U_1, \ldots, U_k) \twoheadrightarrow (X_1, \ldots, X_k)$ for $k \in [n]$, and in particular, $\bar{\mu} \models \mathcal{U} \twoheadrightarrow \mathcal{X}$.

With this in mind, we now return to proving that $\mu$ has the required independencies. It suffices to show that $\mu(\mathcal{X}) = \prod_{i=1}^n \mu(X_i \mid S_i)$. We do so by showing that, for all $k \in [n]$, $\mu(X_1, \ldots, X_k) = \mu(X_1, \ldots, X_{k-1})\mu(X_k \mid S_k)$. By QIM-compatibility witness condition (c), we know that $\bar{\mu} \models (S_k, U_k) \twoheadrightarrow X_k$, and so there exists a function $f_k : V(S_k) \times V(U_k) \to V(X_k)$ for which the event $f_k(S_k, U_k) = X_k$ occurs with probability 1. Since $\bar{\mu} \models (U_1, \ldots, U_{k-1}) \twoheadrightarrow (X_1, \ldots, X_{k-1})$, and $U_k$ is independent of $(U_1, \ldots, U_{k-1})$, it follows from Lemma 10 that $\bar{\mu} \models (X_1, \ldots, X_{k-1}) \perp\!\!\!\perp U_k$. Thus

$$\mu(X_1, \ldots, X_{k-1}, X_k) = \sum_{u \in V(U_k)} \mu(X_1, \ldots, X_{k-1}) \bar{\mu}(U_k = u) \cdot \mathbb{1}[X_k = f_k(S_k, u)]$$

$$= \mu(X_1, \ldots, X_{k-1}) \sum_{u \in V(U_k)} \bar{\mu}(U_k = u) \cdot \mathbb{1}[X_k = f_k(S_k, u)].$$

Observe that the quantity on the right, including the sum, is a function of $X_k$ and $S_k$, but no other variables; let $\varphi(X_k, S_k)$ denote this quantity. Because $\mu$ is a probability distribution, know that $\varphi(X_k, S_k)$ must be the conditional probability of $X_k$ given $X_1, \ldots, X_{k-1}$, and it depends only on the variables $S_k$. Thus $\mu(X_1, \ldots, X_k) = \mu(X_1, \ldots, X_{k-1})\mu(X_k \mid S_k)$.

Therefore $\nu(\mathcal{X}) = \mu(\mathcal{X})$ factors as required by the BN $G$, meaning that $\mu$ has the independencies specified by $G$. (See Koller & Friedman Thm 3.2, for instance.)

( $\impliedby$ ). Suppose $\mu$ satiesfies the independencies of $G$, meaning that each node is conditionally independent of its non-descendents given its parents. We now repeatedly apply the construction Appendix A.1 to construct a QIM-compatibility witness. Specifically, for $k \in \{1, \ldots, n\}$, let $U_k$ be a variable whose values $V(U_k) := V(X_k)^{V(S_k)}$ are functions from values of $X_k$'s parents, to values of $X_k$. Let $\mathcal{U}$ denote the joint variable $(U_1, \ldots, U_n)$, and observe that a setting $\mathbf{g} = (g_1, \ldots, g_n)$ of $\mathcal{U}$ uniquely picks out a value of $\mathcal{X}$, by evaluating each function in order. Let's call this function $f : V(\mathcal{U}) \to V(\mathcal{X})$.

To be more precise, we now construct $f(\mathbf{g})$ inductively. The first component we must produce is $X_1$, but since $X_1$ has no parents, $g_1$ effectively describes a single value of $X_1$, so we define the first component $f(\mathbf{g})[X_1]$ to be that value. More generally, assuming that we have already defined the components $X_1, \ldots, X_{i-1}$, among which are the variables $S_k$ on which $X_i$ depends, we can determine the value of $X_i$; formally, this means defining

$$f(\mathbf{g})[X_i] := g_i(f(\mathbf{g})[S_i]),$$

which, by our inductive assumption, is well-defined. Note that, for all $\mathbf{g} \in V(\mathcal{U})$ and $\mathbf{x} \in V(\mathcal{X})$, the function $f$ is characterized by the property

$$f(\mathbf{g}) = \mathbf{x} \quad \iff \quad \bigwedge_{i=1}^n g_i(\mathbf{x}[S_i]) = \mathbf{x}[X_i]. \tag{4}$$

To quickly verify this: if $f(\mathbf{g}) = \mathbf{x}$, then in particular, for $i \in [n]$, then $\mathbf{x}[X_i] = f(\mathbf{g})[X_i] = g_i(\mathbf{x}[S_i])$ by the definition above. Conversely, if the right hand side of (4) holds, then we can prove $f(\mathbf{g}) = \mathbf{x}$ by induction over our construction of $f$: if $f(\mathbf{g})[X_j] = \mathbf{x}[X_j]$ for all $j < i$, then $f(\mathbf{g})[X_i] = g_i(f(\mathbf{g})[S_i]) = g_i(\mathbf{x}[S_i]) = \mathbf{x}[X_i]$.

Next, we define an unconditional probability over each $U_k$ according to

$$\bar{\mu}_i(U_i = g) := \prod_{\mathbf{s} \in V(S_k)} \mu(X_i = g(s) \mid S_i = \mathbf{s}),$$

which, as verified in Appendix A.1, is indeed a conditional probability, and has the property that $\bar{\mu}_i(U_i(\mathbf{s}) = x) = \mu(X_i = x \mid S_i = \mathbf{s})$ for all $x \in V(X_i)$ and $\mathbf{s} \in V(S_i)$. By taking an independent

combination (tensor product) of each of these unconditional distributions, we obtain a joint distribution $\bar{\mu}(\mathcal{U}) = \prod_{i=1}^{n} \bar{\mu}_i(U_i)$. Finally, we extend this distribution to a full joint distribution $\bar{\mu}(\mathcal{U}, \mathcal{X})$ via the pushforward of $\bar{\mu}(\mathcal{U})$ through the function $f$ defined by induction above. In this distribution, each $X_i$ is determined by $U_i$ and $S_i$.

By construction, the variables $\mathcal{U}$ are mutually independent (for Definition 2(b)), and satisfy $(S_k, U_k) \twoheadrightarrow X_k$ for all $k \in [n]$ (Definition 2(c)). It remains only to verify that the marginal of $\bar{\mu}$ on the variables $\mathcal{X}$ is the original distribution $\mu$ (Definition 2(a)). Here is where we rely on the fact that $\mu$ satisfies the independencies of $G$, which means that we can factor $\mu(\mathcal{X})$ as $\mu(\mathcal{X}) = \prod_{i=1}^{n} \mu(X_i \mid S_i)$.

$$
\begin{aligned}
\bar{\mu}(\mathcal{X}{=}\mathbf{x}) &= \sum_{\mathbf{g} \in V(\mathcal{U})} \bar{\mu}(\mathcal{U}{=}\mathbf{g}) \cdot \delta f(\mathbf{x} \mid \mathbf{g}) \\
&= \sum_{(g_1,\ldots,g_n) \in V(\mathcal{U})} \mathbb{1}\big[\mathbf{x} = f(\mathbf{g})\big] \prod_{i=1}^{n} \bar{\mu}(U_i{=}g_i) \\
&= \sum_{(g_1,\ldots,g_n) \in V(\mathcal{U})} \mathbb{1}\Big[\bigwedge_{i=1}^{n} g_i(\mathbf{x}[S_i]) = \mathbf{x}[X_i]\Big] \prod_{i=1}^{n} \bar{\mu}(U_i{=}g_i) \qquad [\text{ by (4) }] \\
&= \prod_{i=1}^{n} \sum_{g \in V(U_i)} \mathbb{1}\big[g(\mathbf{x}[S_i]) = \mathbf{x}[X_i]\big] \cdot \bar{\mu}(U_i = g) \\
&= \prod_{i=1}^{n} \bar{\mu}\Big(\big\{g \in V(U_i) \,\big|\, g(\mathbf{s}_i) = x_i\big\}\Big) \quad \text{where } \begin{array}{l} x_i := \mathbf{x}[X_i], \\ \mathbf{s}_i := \mathbf{x}[S_i] \end{array} \\
&= \prod_{i=1}^{n} \bar{\mu}\big(U_i(\mathbf{s}_i){=}x_i\big) \\
&= \prod_{i=1}^{n} \mu(X_i = x_i \mid S_i = \mathbf{s}_i) \\
&= \mu(\mathcal{X} = \mathbf{x}).
\end{aligned}
$$

Therefore, when $\mu$ satisfies the independencies of a BN $G$, it is QIM-compatible with $\mathcal{A}_G$. $\qquad \square$

Before we move on to proving the other results in the paper, we first illustrate how this relatively substantial first half of the proof of Theorem 1 can be dramatically simplified by relying on two information theoretic arguments.

*Alternate, information-based proof.* ( $\implies$ ). Let $G$ be a dag. If $\mu \models \Diamond \mathcal{A}_G$, then by Theorem 7, $IDef_{\mathcal{A}_G}(\mu) \leq 0$. In the appendix of [23], it is shown that $IDef_{\mathcal{A}_G}(\mu) \geq 0$ with equality iff $\mu$ satisfies the BN's independencies. Thus $\mu$ must satisfy the appropriate independencies. $\qquad \square$

## Theorem 2.

    (a) $\mu \models X {\twoheadrightarrow} Y \wedge \Diamond \mathcal{A}$   *if and only if*   $\forall n \geq 0.\ \mu \models \Diamond \mathcal{A} \sqcup_{X \to Y}^{(+n)}$ .

    (b) *if* $\mathcal{A} = \mathcal{A}_G$ *for a dag* $G$, *then* $\mu \models X {\twoheadrightarrow} Y \wedge \Diamond \mathcal{A}$ *if and only if* $\mu \models \Diamond \mathcal{A} \sqcup_{X \to Y}^{(+1)}$.

    (c) *if* $\exists a \in \mathcal{A}$ *such that* $S_a = \emptyset$ *and* $X \in T_a$, *then* $\mu \models X {\twoheadrightarrow} Y \wedge \Diamond \mathcal{A}$ *iff* $\mu \models \Diamond \mathcal{A} \sqcup_{X \to Y}^{(+2)}$.

*Proof.* **(a).** The forward direction is straightforward. Suppose that $\mu \models \Diamond \mathcal{A}$ and $\mu \models X \twoheadrightarrow Y$. The former condition gives us a witness $\nu(\mathcal{X}, \mathcal{U})$ in which $\mathcal{U} = \{U_a\}_{a \in \mathcal{A}}$ are mutually independent variables indexed by $\mathcal{A}$, that determine their respective edges. "Extend" $\nu$ in the unique way to $n$ additional constant variables $U_1, \ldots, U_n$, each of which can only take on one value. We claim that this "extended" distribution $\nu'$, which we conflate with $\nu$ because it is not meaningfully different, is a witness to $\mu \models \Diamond \mathcal{A} \sqcup_{X \to Y}^{(+n)}$. Since $\mu \models X \twoheadrightarrow Y$ it must also be that $\nu \models X \twoheadrightarrow Y$, and it follows that $\nu \models (X, U_i) \twoheadrightarrow Y$ for all $i \in \{1, \ldots, n\}$, demonstrating that the new requirements of

$\nu'$ imposed by Definition 2(c) hold. (The remainder of the requirements for condition (c), namely that $\nu' \models (S_a, U_a) \twoheadrightarrow T_a$ for $a \in \mathcal{A}$, still hold because $\nu'$ is an extension of $\nu$, which we know has this property.) Finally, since $\mathcal{U}$ are mutually independent and each $U_i$ is a constant (and hence independent of everything), the variables $\mathcal{U}' := \mathcal{U} \sqcup \{U_i\}_{i=1}^n$ are also mutually independent. Thus $\nu$ (or, more precisely, an isomorphic "extension" of it to additional trivial variables) is a witness of $\mu \models \Diamond \mathcal{A} \sqcup_{X \to Y}^{(+n)}$.

The reverse direction is difficult to prove directly, yet it is a straightforward application of Theorem 7. Suppose that $\mu \models \Diamond \mathcal{A} \sqcup_{X \to Y}^{(+n)}$ for all $n \geq 0$. By Theorem 7, we know that

$$0 \geq IDef_{\mathcal{A} \sqcup_{X \to Y}^{(+n)}}(\mu) = IDef_{\mathcal{A}}(\mu) + n\, \mathrm{H}_\mu(Y|X).$$

Because $IDef_{\mathcal{A}}(\mu)$ is bounded below (by $-\log|\mathrm{V}(\mathcal{X})|$), it cannot be the case that $\mathrm{H}_\mu(Y|X) > 0$; otherwise, the inequality above would not hold for large $n$ (specifically, for $n > \log|\mathrm{V}(\mathcal{X})|/\mathrm{H}_\mu(Y|X)$). By Gibbs inequailty, $\mathrm{H}_\mu(Y|X)$ is non-negative, and thus it must be the case that $\mathrm{H}_\mu(Y|X) = 0$. Thus $\mu \models X \twoheadrightarrow Y$. It is also true that $\mu \models \Diamond \mathcal{A}$ by monotonicity (Proposition 13), which is itself a direct application of Theorem 7

**(b).** Now $\mathcal{A} = \mathcal{A}_G$ for some graph $G$. The forward direction of the equivalence is strictly weaker than the one we already proved in part (a); we have shown $\mu \models \Diamond \mathcal{A} \sqcup_{X \to Y}^{(+n)}$ for all $n \geq 0$, and needed only to show it for $n = 1$. The reverse direction is what's interesting. As before, we will take a significant shortcut by using Theorem 7. Suppose $\mu \models \Diamond \mathcal{A} \sqcup_{X \to Y}^{(+1)}$. In this case where $\mathcal{A} = \mathcal{A}_G$, it was shown by Richardson and Halpern [23] that $IDef_{\mathcal{A}}(\mu) \geq 0$. It follows that

$$0 \overset{\text{(Theorem 7)}}{\geq} IDef_{\mathcal{A} \sqcup_{X \to Y}^{(+n)}}(\mu) = IDef_{\mathcal{A}}(\mu) + \mathrm{H}_\mu(Y|X) \geq 0,$$

and thus $\mathrm{H}_\mu(Y|X) = 0$, meaning that $\mu \models X \twoheadrightarrow Y$ as promised. As before, we also have $\mu \models \Diamond \mathcal{A}$ by monotonicity.

**(c).** As in part (b), the forward direction is a special case of the forward direction of part (a), and it remains only to prove the reverse direction. Equipped with the additional information that $\mathcal{A} \rightsquigarrow \{\to \{X\}\}$, suppose that $\mu \models \Diamond \mathcal{A} \sqcup_{X \to Y}^{(+2)}$. By monotonicity, this means $\mu \models \Diamond \mathcal{A}$ and also that $\mu \models \to \boxed{X} \overset{\rightarrow}{\rightarrow} \boxed{Y}$. Let $\mathcal{A}'$ denote this hypergraph. Once again by appeal to Theorem 7, we have that

$$0 \geq IDef_{\mathcal{A}'} = -\mathrm{H}_\mu(X,Y) + \mathrm{H}(X) + 2\,\mathrm{H}_\mu(Y|X) = \mathrm{H}_\mu(Y|X) \geq 0.$$

It follows that $\mathrm{H}_\mu(Y|X) = 0$, and thus $\mu \models X \twoheadrightarrow Y$. As mentioned above, we also know that $\mu \models \Diamond \mathcal{A}$, and thus $\mu \models \Diamond \mathcal{A} \wedge X \twoheadrightarrow Y$ as promised. $\qquad\square$

## A.3 Causality Results of Section 3

**Proposition 3.** *Given a graph $G$ and a distribution $\mu$, $\mu \models \Diamond \mathcal{A}_G$ iff there exists a fully randomized PSEM of dependency structure $\mathcal{A}_G$ from which $\mu$ can arise.*

*Proof.* ( $\Longrightarrow$ ). Suppose $\mu \models \Diamond \mathcal{A}_G$. Thus there exists some witness $\bar{\mu}(\mathcal{X}, \mathcal{U})$ to this fact, satisfying conditions (a-c) of Definition 2. Because $\mathcal{A}_G$ is partitional, the elements of $\text{PSEMs}_{\mathcal{A}_G}(\bar{\mu})$ are ordinary (i.e., not generalized) randomized PSEMs. We claim that every $\mathcal{M} = (M, P) \in \text{PSEMs}_{\mathcal{A}_G}(\bar{\mu})$ that is a randomized PSEM from which $\mu$ can arise, and also has the property that $\mathbf{Pa}_M(Y) \subseteq \mathbf{Pa}_G(Y) \cup \{U_Y\}$ for all $Y \in \mathcal{X}$.

- The hyperarcs of $\mathcal{A}_G$ correspond to the vertices of $G$, which in turn correspond to the variables in $\mathcal{X}$; thus $\mathcal{U} = \{U_X\}_{X \in \mathcal{X}}$. By property (b) of QIM-compatibility witnesses (Definition 2), these variables $\{U_X\}_{X \in \mathcal{X}}$ are mutually independent according to $\bar{\mu}$. Furthermore, because $\mathcal{M} = (M, P) \in \text{PSEMs}_{\mathcal{A}_G}(\bar{\mu})$, we know that $\bar{\mu}(\mathcal{U}) = P$, and thus the variables in $\mathcal{U}$ must be mutually independent according to $P$. By construction, in causal models $\mathcal{M} \in \text{PSEMs}_{\mathcal{A}_G}(\bar{\mu})$ the equation $f_Y$ can depend only on $S_Y = \mathbf{Pa}_G(Y) \subseteq \mathcal{X}$ and $U_Y$. So, in particular, $f_Y$ does not depend on $U_X$ for $X \neq Y$.

  Altogether, we have shown that $\mathcal{M}$ contains exogenous variables $\{U_X\}_{X \in \mathcal{X}}$ that are mutually independent according to $P$, and that $f_Y$ does not depend on $U_X$ when $X \neq Y$. Thus, $\mathcal{M}$ is a randomized PSEM.

- By condition (a) on QIM-compatibility witnesses (Definition 2), we know that $\bar{\mu}(\mathcal{X}) = \mu$. By Proposition 5(a), we know that $\mu \in \{\!\!\{\mathcal{M}\}\!\!\}$. Together, the previous two sentences mean that $\mu$ can arise from $\mathcal{M}$.

- Finally, as mentioned in the first bullet item, the equation $f_Y$ in $M$ can depend only on $S_Y = \mathbf{Pa}_G(Y)$ and on $U_Y$. Thus $\mathbf{Pa}_M(Y) \subseteq \mathbf{Pa}_G(Y) \cup \{U_Y\}$ for all $Y \in \mathcal{X}$.

Under the assumption that $\mu \models \Diamond \mathcal{A}_G$, we have now shown that there exists a randomized causal model $\mathcal{M}$ from which $\mu$ can arise, with the property that $\mathbf{Pa}_{\mathcal{M}}(Y) \subseteq \mathbf{Pa}_G(Y) \cup \{U_Y\}$ for all $Y \in \mathcal{X}$.

( $\Longleftarrow$ ). Conversely, suppose there is a randomized PSEM $\mathcal{M} = (M = (\mathcal{Y}, \mathcal{U}, \mathcal{F}), P)$ with the property that $\mathbf{Pa}_M(Y) \subseteq \mathbf{Pa}_G(Y) \cup \{U_Y\}$ for all $Y$, from which $\mu$ can arise. The last clause means there exists some $\nu \in \{\!\!\{\mathcal{M}\}\!\!\}$ such that $\nu(\mathcal{X}) = \mu$. We claim that this $\nu$ is a witness to $\mu \models \Diamond \mathcal{A}_G$. We already know that condition (a) of being a QIM-compatibility witness is satisfied, since $\nu(\mathcal{X}) = \mu$. Condition (b) holds because of the assumption that $\{U_X\}_{X \in \mathcal{X}}$ are mutually independent in the distribution $P$ for a randomized PSEM (and the fact that $\nu(\mathcal{U}) = P$, since $\nu \in \{\!\!\{\mathcal{M}\}\!\!\}$). Finally, we must show that (c) for each $Y \in \mathcal{X}$, $\nu \models \mathbf{Pa}_G(Y) \cup \{U_Y\} \twoheadrightarrow Y$. Since $\nu \in \{\!\!\{\mathcal{M}\}\!\!\}$, we know that $M$'s equation holds with probability 1 in $\nu$, and so it must be the case that $\nu \models \mathbf{Pa}_M(Y) \twoheadrightarrow Y$. Note that, in general, if $\mathbf{A} \subseteq \mathbf{B}$ and $\mathbf{A} \twoheadrightarrow \mathbf{C}$, then $\mathbf{B} \twoheadrightarrow \mathbf{C}$. By assumption, $\mathbf{Pa}_M(Y) \subseteq \mathbf{Pa}_G(Y) \cup \{U_Y\}$, and thus $\nu \models \mathbf{Pa}_G(Y) \cup \{U_Y\} \twoheadrightarrow Y$.

Thus $\nu$ satisfies all conditions (a-c) for a QIM-compatibility witness, and hence $\mu \models \Diamond \mathcal{A}_G$. $\qquad \square$

**Proposition 4.** *$\mu \models \Diamond \mathcal{A}$ iff there exists a generalized randomized PSEM with structure $\mathcal{A}$ from which $\mu$ can arise.*

*Proof.* ( $\Longrightarrow$ ). Suppose $\mu \models \Diamond \mathcal{A}$, meaning there exists a witness $\nu(\mathcal{X}, \mathcal{U})$ with property Definition 2(c), meaning that, for all $a \in \mathcal{A}$, there is a functional dependence $(S_a, U_a) \twoheadrightarrow T_a$. Thus, there is some set of functions $\mathcal{F}$ with these types that holds with probability 1 according to $\nu$. Meanwhile, by Definition 2(b), $\nu(\mathcal{U})$ are mutually independent, so defining $P_a(U_a) := \nu(U_a)$, we have $\nu(\mathcal{U}) = \prod_{a \in \mathcal{A}} P_a(U_a)$. Together, the previous two conditions (non-deterministically) define a generalized randomized PSEM $\mathcal{M}$ of shape $\mathcal{A}$ for which $\nu \in \{\!\!\{\mathcal{M}\}\!\!\}$. Finally, by Definition 2(a), we know that $\mu$ can arise from $\mathcal{M}$.

( $\Longleftarrow$ ). Conversely, suppose there is a generalized randomized SEM $\mathcal{M}$ of shape $\mathcal{A}$ from which $\mu(\mathcal{X})$ can arise. Thus, there is some $\nu \in \{\!\!\{\mathcal{M}\}\!\!\}$ whose marginal on $\mathcal{X}$ is $\mu$. We claim that this $\nu$ is also

a witness that $\mu \models \Diamond \mathcal{A}$. The marginal constraint from Definition 2(a) is clearly satisfied. Condition (b) is immediate as well, because $\nu(\mathcal{U}) = \prod_a P_a(U_a)$. Finally, condition (c) is satisfied, because the equations of $\mathcal{M}$ hold with probability 1, ensuring the appropriate functional dependencies. $\square$

**Proposition 5.** *If $\bar{\mu}(\mathcal{X}, \mathcal{U}_\mathcal{A})$ is a witness for QIM-compatibility with $\mathcal{A}$ and $\mathcal{M}$ is a PSEM with dependency structure $\mathcal{A}$, then $\bar{\mu} \in \{\!\{\mathcal{M}\}\!\}$ if and only if $\mathcal{M} \in \mathrm{PSEMs}_\mathcal{A}(\bar{\mu})$.*

*Proof.* (a) is straightforward. Suppose $\mathcal{M} \in \mathrm{PSEMs}(\nu)$. By construction, the equations of $\mathcal{M}$ reflect functional dependencies in $\nu$, and hence hold with probability 1.[6] Furthermore, the distribution $P(\mathcal{U})$ in all $\mathcal{M} \in \mathrm{PSEMs}(\nu)$ is equal to $\nu(\mathcal{U})$. These two facts, demonstrate that $\nu$ satisfies the two constraints required for membership in $\{\!\{\mathcal{M}\}\!\}$.

(b). We do the two directions separately. First, suppose $\mathcal{M} \in \mathrm{PSEMs}(\nu)$. We have already shown (in part (a)) that $\nu \in \{\!\{\mathcal{M}\}\!\}$. The construction of $\mathrm{PSEMs}(\nu)$ depends on the hypergraph $\mathcal{A}$ (even if the dependence is not explicitly clear from our notation) in such a way that $f_X$ does not depend on any variables beyond $U_a$ and $S_{a_X}$. Thus, $\mathbf{Pa}_\mathcal{M}(X) \subseteq S_{a_X} \cup \{U_{a_X}\}$.

Conversely, suppose $\mathcal{M} = (\mathcal{X}, \mathcal{U}, \mathcal{F})$ is a PSEM satisfying $\nu \in \{\!\{\mathcal{M}\}\!\}$ and $\mathbf{Pa}_\mathcal{M}(X) \subseteq S_{a_X} \cup \{U_{a_X}\}$. We would like to show that $\mathcal{M} \in \mathrm{PSEMs}(\nu)$. Because $\nu \in \{\!\{\mathcal{M}\}\!\}$, we know that the distribution $P(\mathcal{U})$ over the exogenous variables in the PSEM $\mathcal{M}$ is equal to $\nu(\mathcal{U})$, matching the first part of our construction. What remains is to show that the equations $\mathcal{F}$ are consistent with our transformation. Choose any $X \in \mathcal{X}$. Because $\mathcal{A}$ is subpartitional, there is a unique $a_X \in \mathcal{A}$ such that $X \in T_{a_X}$. Now choose any values $\mathbf{s} \in \mathrm{V}(S_{a_X})$ and $u \in \mathrm{V}(U_{a_X})$. If $\nu(\mathbf{s}, u) > 0$, then we know there is a unique value of $x \in \mathrm{V}(X)$ such that $\nu(\mathbf{s}, u, x) > 0$. Since $\mathcal{M}$'s equation for $X$, $f_X$, depends only on $\mathbf{s}$ and $u$, and holds with probability 1, we know that $f_X(\mathbf{s}, u) = t$, as required. On the other hand, if $\nu(\mathbf{s}, u) = 0$, then any choice of $f_X(\mathbf{s}, u)$ is consistent with our procedure. Since this is true for all $X$, and all possible inputs to the equation $f_X$, we conclude that the equations $\mathcal{F}$ can arise from the procedure described in the main text, and therefore $\mathcal{M} \in \mathrm{PSEMs}(\nu)$. $\square$

**Theorem 6.** *Suppose that $\bar{\mu}$ is a QIM witness for $\mathcal{A}$, that $\mathcal{M} = (\mathcal{U}, \mathcal{V}, \mathcal{F}, P) \in \mathrm{PSEMs}_\mathcal{A}(\bar{\mu})$ is a corresponding PSEM, and that the noise variables $\mathcal{U}_\mathcal{A} = \{U_X\}_{X \in \mathcal{V}}$ are independent of the other exogenous variables $\mathcal{U} \setminus \mathcal{U}_\mathcal{A}$. For all $\mathbf{X} \subseteq \mathcal{V}$ and $\mathbf{x} \in \mathrm{V}(\mathbf{X})$, if $\bar{\mu}(\mathrm{do}_\mathcal{M}(\mathbf{X}=\mathbf{x})) > 0$, then*

- *(a) $\bar{\mu}(\mathcal{V} \mid \mathrm{do}_\mathcal{M}(\mathbf{X}=\mathbf{x}))$ can arise from $\mathcal{M}_{\mathbf{X}\leftarrow\mathbf{x}}$;*
- *(b) for all events $\varphi \subseteq \mathrm{V}(\mathcal{V})$, $\mathrm{Pr}_\mathcal{M}\left([\mathbf{X}\leftarrow\mathbf{x}]\varphi\right) \leq \bar{\mu}\left(\varphi \mid \mathrm{do}_\mathcal{M}(\mathbf{X}=\mathbf{x})\right) \leq \mathrm{Pr}_\mathcal{M}\left(\langle\mathbf{X}\leftarrow\mathbf{x}\rangle\varphi\right)$ and all three are equal when $\mathcal{M} \models \mathcal{U} \twoheadrightarrow \mathcal{V}$ (such as when $\mathcal{M}$ is acyclic).*

*Proof.* **(part a).** Because we have assumed $\bar{\mu}(\mathrm{do}_\mathcal{M}(\mathbf{X}=\mathbf{x})) > 0$, the conditional distribution

$$\bar{\mu} \mid \mathrm{do}_\mathcal{M}(\mathbf{X}=\mathbf{x}) = \bar{\mu}(\mathcal{U}, \mathcal{X}) \cdot \prod_{X \in \mathbf{X}} \mathbb{1}\left[\forall \mathbf{s}. f_X(U_X, \mathbf{s}) = \mathbf{x}[X]\right] \Big/ \bar{\mu}(\mathrm{do}_\mathcal{M}(\mathbf{X}=\mathbf{x}))$$

is defined. By assumption, $\mathcal{M} \in \mathrm{PSEMs}(\bar{\mu})$ and $\bar{\mu}$ is a witness to the fact that $\mu \models \Diamond \mathcal{A}$. Thus, by Proposition 5, we know that $\bar{\mu} \in \{\!\{\mathcal{M}\}\!\}$. So in particular, all equations of $\mathcal{M}$ hold for all joint settings $(\mathbf{u}, \mathbf{v}) \in \mathrm{V}(\mathcal{U} \cup \mathcal{V})$ in the support of $\bar{\mu}$. But the support of the conditional distribution $\bar{\mu} \mid \mathrm{do}_\mathcal{M}(\mathbf{X}=\mathbf{x})$ is a subset of the support of $\bar{\mu}$, so all equations of $\mathcal{M}$ also hold in the conditioned distribution. Furthermore, the event $\mathrm{do}_\mathcal{M}(\mathbf{X}=\mathbf{x})$ is the event in which, for all $X \in \mathbf{X}$, the variable $U_X$ takes on a value such that $f_X(\ldots, U_X, \ldots) = \mathbf{x}[X]$. Thus the equations corresponding to $\mathbf{X} = \mathbf{x}$ also hold with probability 1 in $\bar{\mu} \mid \mathrm{do}_\mathcal{M}(\mathbf{X}=\mathbf{x})$.

This shows that all equations of $\mathcal{M}_{\mathbf{X}\leftarrow\mathbf{x}}$ hold with probability 1 in $\bar{\mu} \mid \mathrm{do}_\mathcal{M}(\mathbf{X}=\mathbf{x})$. However, the marginal distribution $\bar{\mu}(\mathcal{U} \mid \mathrm{do}_\mathcal{M}(\mathbf{X}=\mathbf{x}))$ over $\mathcal{U}$ is typically not equal to $P(\mathcal{U})$—after all, we have collapsed distribution of the variables $\mathcal{U}_\mathbf{X} := \{U_X : X \in \mathbf{X}\}$. Clearly, $\bar{\mu} \mid \mathrm{do}_\mathcal{M}(\mathbf{X}=\mathbf{x}) \notin \{\!\{\mathcal{M}_{\mathbf{X}\leftarrow\mathbf{x}}\}\!\}$. However, as we now show, there exists a *different* distribution $\bar{\mu}' \in \{\!\{\mathcal{M}_{\mathbf{X}\leftarrow\mathbf{x}}\}\!\}$ such that $\bar{\mu}'(\mathcal{V}) = \nu(\mathcal{V} \mid \mathrm{do}_\mathcal{M}(\mathbf{X}=\mathbf{x}))$.

---

[6]When the probability of some combination of source variables is zero, there is typically more than one choice of functions that holds with probability 1; the choice of functions is essentially the choice of $\mathcal{M} \in \mathrm{PSEMs}(\nu)$.

Let $\mathcal{U}_0 := \mathcal{U} \setminus \mathcal{U}_{\mathbf{X}}$. We can define $\bar{\mu}'$ according to

$$\bar{\mu}'(\mathcal{V}, \mathcal{U}_{\mathbf{X}}, \mathcal{U}_0) := \bar{\mu}(\mathcal{V}, \mathcal{U}_0 \mid \mathrm{do}_{\mathcal{M}}(\mathbf{X}{=}\mathbf{x}))P(\mathcal{U}_{\mathbf{X}}).$$

The distribution $\bar{\mu}'$ satisfies three critical properties:

1. By construction, the marginal of $\bar{\mu}'$ on $\mathcal{V}$ is $\bar{\mu}'(\mathcal{V}) = \bar{\mu}(\mathcal{V} \mid \mathrm{do}_{\mathcal{M}}(\mathbf{X}{=}\mathbf{x}))$.

2. At the same time, the marginal of $\bar{\mu}'$ on exogenous variables is

$$
\begin{aligned}
\bar{\mu}'(\mathcal{U}) &= \bar{\mu}'(\mathcal{U}_{\mathbf{X}}, \mathcal{U}_0) \\
&= \int_{V(\mathcal{V})} \bar{\mu}'(\mathbf{v}, \mathcal{U}_{\mathbf{X}}, \mathcal{U}_0)\,\mathrm{d}\mathbf{v} \\
&= \int_{V(\mathcal{V})} \bar{\mu}(\mathbf{v}, \mathcal{U}_0 \mid \mathrm{do}_{\mathcal{M}}(\mathbf{X}{=}\mathbf{x}))P(\mathcal{U}_{\mathbf{X}})\,\mathrm{d}\mathbf{v} && [\text{definition of } \bar{\mu}'] \\
&= P(\mathcal{U}_{\mathbf{X}})\bar{\mu}(\mathcal{U}_0 \mid \mathrm{do}_{\mathcal{M}}(\mathbf{X}{=}\mathbf{x})) \\
&= P(\mathcal{U}_{\mathbf{X}})P(\mathcal{U}_0 \mid \mathrm{do}_{\mathcal{M}}(\mathbf{X}{=}\mathbf{x})) && [\text{because } \bar{\mu}(\mathcal{U}) = P(\mathcal{U}), \text{ since } \bar{\mu} \in \{\!\{\mathcal{M}\}\!\}] \\
&= P(\mathcal{U}_{\mathbf{X}})P(\mathcal{U}_0) && \begin{bmatrix} \text{since } \mathrm{do}_{\mathcal{M}}(\mathbf{X}{=}\mathbf{x}) \text{ depends only on} \\ \mathcal{U}_{\mathbf{X}}, \text{ while } \mathcal{U}_{\mathbf{X}} \text{ and } \mathcal{U}_{\mathbf{Z}} \text{ are indepen-} \\ \text{dent in } \bar{\mu} \text{ (by the witness condition).} \end{bmatrix} \\
&= P(\mathcal{U}_{\mathbf{X}}, \mathcal{U}_0) && [\text{ same reason as above }]
\end{aligned}
$$

3. Finally, $\bar{\mu}'$ satisfies all equations of $\mathcal{M}_{\mathbf{X}\leftarrow\mathbf{x}}$. It satisfies the equations for the variables $\mathbf{X}$ because $\mathbf{X} = \mathbf{x}$ holds with probability 1. At the same time, the equations of $\mathcal{M}_{\mathbf{X}\leftarrow\mathbf{x}}$ corresponding to other variables, say $f_Z$ for $Z \in \mathcal{V} \setminus \mathbf{X}$, also hold with probability one. This is because the marginal $\bar{\mu}'(\mathcal{U}_{\mathbf{Z}}, \mathcal{V})$ is shared with the distribution $\bar{\mu} \mid \mathrm{do}_{\mathcal{M}}(\mathbf{X}{=}\mathbf{x})$, and that distribution satisfies these equations. (It suffices to show that they share this particular marginal because the equations for $\mathbf{Z}$ do not depend on $\mathcal{U}_{\mathbf{X}}$.)

Together, properties 2 and 3 show that $\bar{\mu}' \in \{\!\{\mathcal{M}_{\mathbf{X}\leftarrow\mathbf{x}}\}\!\}$, while property 1 shows that $\bar{\mu}(\mathcal{V} \mid \mathrm{do}_{\mathcal{M}}(\mathbf{X}{=}\mathbf{x}))$ can arise from $\mathcal{M}_{\mathbf{X}\leftarrow\mathbf{x}}$. This establishes part (a).

**(part b).** We will again make use of the distribution $\bar{\mu}'$ defined in part (a), and its three critical properties listed above. Given a setting $\mathbf{u} \in V(\mathcal{U})$ of the exogenous variables, let

$$\mathcal{F}_{\mathbf{X}\leftarrow\mathbf{x}}(\mathbf{u}) := \left\{ \omega \in V(\mathcal{V}) \;\middle|\; \begin{array}{ll} \forall X \in \mathbf{X}. & \omega[X] = \mathbf{x}[X] \\ \forall Y \in \mathcal{V} \setminus \mathbf{X}. & \omega[Y] = f_X(\omega[\mathcal{X} \setminus Y], \mathbf{u}) \end{array} \right\}$$

denote the set of joint settings of endogenous variables that are consistent with the equations of $\mathcal{M}_{\mathbf{X}\leftarrow\mathbf{x}}$.

If $\mathbf{u} \in V(\mathcal{U})$ is such that

$$
\begin{aligned}
(M, \mathbf{u}) \models [\mathbf{X}{\leftarrow}\mathbf{x}]\varphi \quad &\Longleftrightarrow \quad (M_{\mathbf{X}\leftarrow\mathbf{x}}, \mathbf{u}) \models \varphi \\
&\Longleftrightarrow \quad \forall \omega \in \mathcal{F}_{\mathbf{X}\leftarrow\mathbf{x}}(\mathbf{u}).\ \omega \in \varphi \\
&\Longleftrightarrow \quad \mathcal{F}_{\mathbf{X}\leftarrow\mathbf{x}}(\mathbf{u}) \subseteq \varphi,
\end{aligned}
$$

then $\phi$ holds at all points that satisfy the equations of $M_{\mathbf{X}\leftarrow\mathbf{x}}$. So, since $\bar{\mu}'$ is supported only on such points (property 3), it must be that $\bar{\mu}'(\varphi) = 1$. By property 1, $\bar{\mu}'(\varphi) = \bar{\mu}(\varphi \mid \mathrm{do}_{\mathcal{M}}(\mathbf{X}{=}\mathbf{x}))$.

Furthermore, if $\bar{\mu}'(\varphi) > 0$, then there must exist some $\omega \in \mathcal{F}_{\mathbf{X}\leftarrow\mathbf{x}}(\mathbf{u})$ satisfying $\varphi$, and thus $(M, \mathbf{u}) \models \langle \mathbf{X}{\leftarrow}\mathbf{x}\rangle\varphi$. Putting both of these observations together, and with a bit more care to the

symbolic manipulation, we find that:

$$\Pr_{\mathcal{M}}([\mathbf{X}{\leftarrow}\mathbf{x}]\varphi) = P(\{\mathbf{u} \in \mathrm{V}(\mathcal{U}) : (M,\mathbf{u}) \models [\mathbf{X}{\leftarrow}\mathbf{x}]\varphi\})$$

$$= \sum_{\mathbf{u} \in \mathrm{V}(\mathcal{U})} P(\mathbf{u})\mathbb{1}\big[\mathcal{F}_{\mathbf{X}{\leftarrow}\mathbf{x}}(\mathbf{u}) \subseteq \varphi\big]$$

$$\leq \sum_{\mathbf{u} \in \mathrm{V}(\mathcal{U})} P(\mathbf{u})\bar{\mu}'(\varphi \mid \mathbf{u}) \qquad = \bar{\mu}'(\varphi) = \bar{\mu}(\varphi \mid \mathrm{do}_{\mathcal{M}}(\mathbf{X}{=}\mathbf{x}))$$

$$\leq \sum_{\mathbf{u} \in \mathrm{V}(\mathcal{U})} P(\mathbf{u})\mathbb{1}\big[\mathcal{F}_{\mathbf{X}{\leftarrow}\mathbf{x}}(\mathbf{u}) \cap \varphi \neq \emptyset\big]$$

$$= P(\{\mathbf{u} \in \mathrm{V}(\mathcal{U}) : (M,\mathbf{u}) \models \langle\mathbf{X}{\leftarrow}\mathbf{x}\rangle\varphi\})$$

$$= \Pr_{\mathcal{M}}(\langle\mathbf{X}{\leftarrow}\mathbf{x}\rangle\varphi), \qquad \text{as desired.}$$

Finally, if $\bar{\mu} \models \mathcal{U} \twoheadrightarrow \mathcal{V}$, then $\mathcal{F}_{\mathbf{X}{\leftarrow}\mathbf{x}}(\mathbf{u})$ is a singleton for all $\mathbf{u}$, and hence $\varphi$ holding for all $\omega \in \mathcal{F}_{\mathbf{X}{\leftarrow}\mathbf{x}}$ and for some $\omega \in \mathcal{F}_{\mathbf{X}{\leftarrow}\mathbf{x}}$ are equivalent. So, in this case,

$$(M,\mathbf{u}) \models [\mathbf{X}{\leftarrow}\mathbf{x}]\varphi \qquad \Longleftrightarrow \qquad (M,\mathbf{u}) \models \langle\mathbf{X}{\leftarrow}\mathbf{x}\rangle\varphi,$$

and thus the probability of both formulas are the same—and it must also equal $\bar{\mu}(\varphi \mid \mathrm{do}_{\mathcal{M}}(\mathbf{X}{=}\mathbf{x}))$ which we have shown lies between them. $\square$

### A.4  Information Theoretic Results of Section 4

To prove Theorem 7 and Theorem 9(a), we will need the following Lemma.

**Lemma 12.** *Consider a set of variables* $\mathbf{Y} = \{Y_1, \ldots, Y_n\}$, *and another (set of) variable(s)* $X$. *Every joint distribution* $\mu(X, \mathbf{Y})$ *over the values of* $X$ *and* $\mathbf{Y}$ *satisfies*

$$\sum_{i=1}^{n} \mathrm{I}_\mu(X \,;\, Y_i) \;\leq\; \mathrm{I}_\mu(X \,;\, \mathbf{Y}) + \sum_{i=1}^{n} \mathrm{H}_\mu(Y_i) - \mathrm{H}_\mu(\mathbf{Y}).$$

*Proof.* Since there is only one joint distribution in scope, we omit the subscript $\mu$, writing $\mathrm{I}(-)$ instead of $\mathrm{I}_\mu(-)$ and $\mathrm{H}(-)$ instead of $\mathrm{H}_\mu(-)$, in the body of this proof. The following fact will also be very useful:

$$\mathrm{I}(A; B, C) = \mathrm{I}(A; C) + \mathrm{I}(A; B \mid C) \qquad \text{(the chain rule for mutual information)}. \qquad (5)$$

We prove this by induction on $n$. In the base case ($n = 1$), we must show that $\mathrm{I}(X; Y) \leq \mathrm{I}(X; Y) + \mathrm{H}(Y) - \mathrm{H}(Y)$, which is an obvious tautology. Now, suppose inductively that

$$\sum_{i=1}^{k} \mathrm{I}(X \,;\, Y_i) \;\leq\; \mathrm{I}(X \,;\, \mathbf{Y}_{1:k}) + \sum_{i=1}^{k} \mathrm{H}(Y_i) - \mathrm{H}(\mathbf{Y}_{1:k}) \qquad\qquad (\mathrm{IH}_k)$$

for some $k < n$, where $\mathbf{Y}_{1:k} = (Y_1, \ldots, Y_k)$. We now prove that the analogue for $k + 1$ also holds. Some calculation reveals:

$$\mathrm{I}(X; Y_{k+1})$$
$$= \mathrm{I}(X; \mathbf{Y}_{1:k+1}) - \mathrm{I}(X; \mathbf{Y}_{1:k} \mid Y_{k+1}) \qquad\qquad \big[\text{ by MI chain rule } (5) \big]$$
$$\leq \mathrm{I}(X; \mathbf{Y}_{1:k+1}) \qquad\qquad \Big[\text{ since } \mathrm{I}(X; \mathbf{Y}_{1:k} \mid Y_{k+1}) \geq 0 \Big]$$
$$= \mathrm{I}(X; Y_{k+1} \mid \mathbf{Y}_{1:k}) + \mathrm{I}(\mathbf{Y}_{1:k}; Y_{k+1}) \qquad\qquad \big[\text{ by MI chain rule } (5) \big]$$
$$= \begin{pmatrix} \mathrm{I}(X; \mathbf{Y}_{1:k+1}) + \mathrm{H}(Y_{k+1}) - \mathrm{H}(\mathbf{Y}_{1:k+1}) \\ -\,\mathrm{I}(X; \mathbf{Y}_{1:k}) \qquad\qquad\qquad + \mathrm{H}(\mathbf{Y}_{1:k}) \end{pmatrix} \quad \begin{bmatrix} \text{left: one more MI chain rule } (5); \\ \text{right: defn of mutual information} \end{bmatrix}.$$

Observe: adding this inequality to our inductive hypothesis $(\mathrm{IH}_k)$ yields $(\mathrm{IH}_{k+1})$! So, by induction, the lemma holds for all $k$. $\square$

**Theorem 7.** *If $\mu \models \Diamond\mathcal{A}$, then $IDef_{\mathcal{A}}(\mu) \leq 0$.*

*Proof.* Suppose that $\mu \models \Diamond\mathcal{A}$, meaning that there is a witness $\nu(\mathcal{X}, \mathcal{U})$ that extends $\mu$, and has properties (a-c) of Definition 2. For each hyperarc a, since $\nu \models (S_a, U_a) \twoheadrightarrow T_a$, we have $H_\nu(T_a \mid S_a, U_a) = 0$, and so

$$H_\mu(T_a \mid S_a) = H_\nu(T_a \mid S_a, U_a) + I_\nu(T_a; U_a \mid S_a) = I_\nu(T_a; U_a \mid S_a).$$

Thus, we compute

$$
\begin{aligned}
\sum_{a \in \mathcal{A}} H_\mu(T_a \mid S_a) &= \sum_{a \in \mathcal{A}} I_\nu(U_a; T_a \mid S_a) \\
&= \sum_{a \in \mathcal{A}} I_\nu(U_a; T_a, S_a) - I_\nu(U_a; S_a) && \text{by MI chain rule (5)} \\
&\leq \sum_{a \in \mathcal{A}} I_\nu(U_a; T_a, S_a) && \text{since } I_\nu(U_a \,;\, S_a) \geq 0 \\
&\leq \sum_{a \in \mathcal{A}} I_\nu(U_a; \mathcal{X}) && \text{since } \mathcal{X} \twoheadrightarrow (S_a, T_a) \\
&\leq I_\nu(\mathcal{X}; \mathcal{U}) + \sum_{a \in \mathcal{A}} H_\nu(U_a) - H_\nu(\mathcal{U}) && \text{by Lemma 12} \\
&= I_\nu(\mathcal{X}; \mathcal{U}) && \begin{array}{c}\text{since } \mathcal{U} \text{ are independent} \\ \text{(per condition (b) of Definition 2)}\end{array} \\
&\leq H_\nu(\mathcal{X}) = H_\mu(\mathcal{X}). && \text{(per condition (a) of Definition 2)}
\end{aligned}
$$

Thus, $IDef_{\mathcal{A}}(\mu) \leq 0$. $\qquad\square$

**Proposition 8.** $\text{QIM}Inc_{\mathcal{A}}(\mu) \geq 0$, with equality iff $\mu \models \Diamond\mathcal{A}$.

*Proof.* The first term in the definition of QIM*Inc* be written as

$$\left( - H_\nu(\mathcal{U}) + \sum_{a \in \mathcal{A}} H_\nu(U_a) \right) = \mathbb{E}_\nu\left[ \log \frac{\nu(\mathcal{U})}{\prod_a \nu(U_a)} \right]$$

and is therefore the relative entropy between $\nu(\mathcal{U})$ and the independent product distribution $\prod_{a \in \mathcal{A}} \nu(U_a)$. Thus, it is non-negative. The remaining terms of $\text{QIM}Inc_{\mathcal{A}}(\mu)$, are all conditional entropies, and hence non-negative as well. Thus $\text{QIM}Inc_{\mathcal{A}}(\mu) \geq 0$.

Now, suppose $\mu$ is s2-comaptible with $\mathcal{A}$, i.e., there exists some $\nu(\mathcal{U}, \mathcal{X})$ such that (a) $\nu(\mathcal{X}) = \mu(\mathcal{X})$, (b) $H_\nu(T_a|S_a, U_a) = 0$, and (d) $\{U_a\}_{a \in \mathcal{A}}$ are mutually independent. Then clearly $\nu$ satisfies the condition under the infemum, every $H_\nu(T_a|S_a, U_a)$ is zero. It is also immediate that the final term is zero as well, because it equals $D(\nu(\mathcal{U}) \parallel \prod_a \nu(U_a))$, and $\nu(\mathcal{U}) = \prod_a \nu(U_a)$, per the definition of mutual independence. Thus, $\nu$ witnesses that $\text{QIM}Inc_{(\mathcal{A}, \lambda)} = 0$.

Conversely, suppose $\text{QIM}Inc_{(\mathcal{A}, \lambda)} = 0$. Because the feasible set is closed and bounded, as is the function, the infemum is achieved by some joint distribution $\nu(\mathcal{X}, \mathcal{A})$ with marginal $\mu(\mathcal{X})$. In this distribution $\nu$, we know that every $H_\nu(T_a|S_a, U_a) = 0$ and $D(\nu(\mathcal{U}) \parallel \prod_a \nu(U_a)) = 0$— because if any of these terms were positive, then the result would be positive as well. So $\nu$ satisfies (a) and (b) by definition. And, because relative entropy is zero iff its arguments are identical we have $\nu(\mathcal{U}) = \prod_a \nu(U_a)$, so the $U_a$'s are mutually independent, and $\nu$ satisfies (d) as well. $\qquad\square$

**Theorem 9.**

(a) *If $(\mathcal{X}, \mathcal{A})$ is a hypergraph, $\mu(\mathcal{X})$ is a distribution, and $\nu(\mathcal{X}, \mathcal{U})$ is an extension of $\nu$ to additional variables $\mathcal{U} = \{U_a\}_{a \in \mathcal{A}}$ indexed by $\mathcal{A}$, then:*

$$IDef_{\mathcal{A}}(\mu) \leq \text{QIM}Inc_{\mathcal{A}}(\mu) \leq IDef_{\mathcal{A}^\dagger}(\nu).$$

(b) *For all $\mu$ and $\mathcal{A}$, there is a choice of $\nu$ that achieves the upper bound. That is,*

$$\mathrm{QIM}Inc_{\mathcal{A}}(\mu) = \min \left\{ IDef_{\mathcal{A}^\dagger}(\nu) : \begin{array}{c} \nu \in \Delta\mathrm{V}(\mathcal{X},\mathcal{U}) \\ \nu(\mathcal{X}) = \mu(\mathcal{X}) \end{array} \right\}.$$

*Proof.* Part (a). The left hand side of the theorem ($IDef_{\mathcal{A}}(\nu) \leq \mathrm{QIM}Inc_{\mathcal{A}}(\mu)$) is a strengthening of the argument used to prove Theorem 7. Specifically, let $\nu^*$ be a minimizer of the optimization problem defining QIM*Inc* We calculate

$\mathrm{QIM}Inc_{\mathcal{A}}(\mu) - IDef_{\mathcal{A}}(\mu)$

$$= \left( \sum_{a \in \mathcal{A}} \mathrm{H}_{\nu^*}(T_a \mid S_a, U_a) - \mathrm{H}_{\nu^*}(\mathcal{U}) + \sum_{a \in \mathcal{A}} \mathrm{H}_{\nu^*}(U_a) \right) - \left( \sum_{a \in \mathcal{A}} \mathrm{H}_\mu(T_a \mid S_a) - \mathrm{H}_\mu(\mathcal{X}) \right)$$

$$= \sum_{a \in \mathcal{A}} \left( \mathrm{H}_{\nu^*}(T_a \mid S_a, U_a) - \mathrm{H}_{\nu^*}(T_a \mid S_a) \right) + \mathrm{H}_\mu(\mathcal{X}) - \mathrm{H}_{\nu^*}(\mathcal{U}) + \sum_{a \in \mathcal{A}} \mathrm{H}_{\nu^*}(U_a)$$

$$= - \sum_{a \in \mathcal{A}} \mathrm{I}_{\nu^*}(T_a; U_a \mid S_a) \quad + \mathrm{H}_\mu(\mathcal{X}) - \mathrm{H}_{\nu^*}(\mathcal{U}) + \sum_{a \in \mathcal{A}} \mathrm{H}_{\nu^*}(U_a).$$

The argument given in the first five lines of the proof of Theorem 7, gives us a particularly convenient bound for the first group of terms on the left:

$$\sum_{a \in \mathcal{A}} \mathrm{I}_{\nu^*}(U_a; T_a \mid S_a) \leq \mathrm{I}_{\nu^*}(\mathcal{X}; \mathcal{U}) + \sum_{a \in \mathcal{A}} \mathrm{H}_{\nu^*}(U_a) - \mathrm{H}_{\nu^*}(\mathcal{U}).$$

Substituting this into our previous expression, we have:

$\mathrm{QIM}Inc_{\mathcal{A}}(\mu) - IDef_{\mathcal{A}}(\mu)$

$$\geq - \left( \mathrm{I}_{\nu^*}(\mathcal{X};\mathcal{U}) + \sum_{a \in \mathcal{A}} \mathrm{H}_{\nu^*}(U_a) - \mathrm{H}_{\nu^*}(\mathcal{U}) \right) + \mathrm{H}_\mu(\mathcal{X}) - \mathrm{H}_{\nu^*}(\mathcal{U}) + \sum_{a \in \mathcal{A}} \mathrm{H}_{\nu^*}(U_a)$$

$$= \mathrm{H}_\mu(\mathcal{X}) - \mathrm{I}_{\nu^*}(\mathcal{X};\mathcal{U})$$

$$\geq 0.$$

The final inequality holds because of our assumption that the marginal $\nu^*(\mathcal{X})$ equals $\mu(\mathcal{X})$. Thus, $\mathrm{QIM}Inc_{\mathcal{A}}(\mu) \geq IDef_{\mathcal{A}}(\mu)$, as proimised.

We now turn to the right hand inequality, and part (b) of the theorem. Recall that $\nu^*$ is defined to be a minimizer of the optimization problem defining QIM*Inc*. For the right inequality ($\mathrm{QIM}Inc_{\mathcal{A}}(\mu) \leq IDef_{\mathcal{A}^\dagger}(\nu)$) of part (a), observe that

$$IDef_{\mathcal{A}^\dagger}(\nu) = - \mathrm{H}_\nu(\mathcal{X},\mathcal{U}) + \sum_{a \in \mathcal{A}} \mathrm{H}_\nu(U_a) + \sum_{a \in \mathcal{A}} \mathrm{H}_\nu(T_a|S_a,U_a) + \mathrm{H}_\nu(\mathcal{X} \mid \mathcal{U})$$

$$= \left( - \mathrm{H}_\nu(\mathcal{U}) + \sum_{a \in \mathcal{A}} \mathrm{H}_\nu(U_a) \right) + \sum_{a \in \mathcal{A}} \mathrm{H}_\nu(T_a|S_a,U_a)$$

$$\geq \left( - \mathrm{H}_{\nu^*}(\mathcal{U}) + \sum_{a \in \mathcal{A}} \mathrm{H}_{\nu^*}(U_a) \right) + \sum_{a \in \mathcal{A}} \mathrm{H}_{\nu^*}(T_a|S_a,U_a)$$

$$= \mathrm{QIM}Inc(\mu).$$

This proves the right hand side of the inequality of part (a). Moreover, because the one inequality holds with equality when $\nu = \nu^*$ is a minimizer of this quantity (subject to having marginal $\mu(\mathcal{X})$) we have shown part (b) as well.

$\square$

## B  Monotonicity and Undirected Graphical Models

The fact that (quantitative) PDG inconsistency is monotonic is a powerful reasoning principle that can be used to prove many important inequalities [22]. In this section, we develop a related principle for QIM-compatibility. Here is a direct but not very useful analague: if $\mathcal{A} \subseteq \mathcal{A}'$ and $\mu \models \Diamond\mathcal{A}'$, conclude $\mu \models \Diamond\mathcal{A}$. After all, if $\mu$ is consistent with a set of independent causal mechanisms, then surely it is consistent with a causal picture in which only a subset of those mechanisms are present and independent. There is a sense in which BNs and MRFs are also monotonic, but in the opposite direction: adding edges to a graph results in a weaker independence statement. We will soon see why.

Since we use *directed* hypergraphs, there is actually a finer notion of monotonicity at play. Inputs and ouputs play opposite roles, and they are naturally monotonic in opposite directions. If there is an obvious way to regard an element of $B$ as an element of $B'$ (abbreviated $B \hookrightarrow B'$), and $A' \hookrightarrow A$, then a function $f : A \to B$ can be regarded as one of type $A' \to B'$. This is depicted to the right. The same principile applies in our setting. If $\mathbf{X}$ and $\mathbf{Z}$ are sets of variables and $\mathbf{X} \subseteq \mathbf{Z}$, then $V(\mathbf{Z}) \hookrightarrow V(\mathbf{X})$, by restriction. It follows, for example, that any mechanism by which $X$ determines $(Y, Y')$ can be viewed as a mechanism by which $(X, X')$ determines $Y$. The general phenomenon is captured by the following.

$$A \xrightarrow{f} B$$
$$\uparrow \quad \updownarrow \quad \downarrow$$
$$A' \dashrightarrow B'$$

**Definition 6.** If $\mathcal{A} = \{S \xrightarrow{a} T\}_a$, $\mathcal{A}' = \{S' \xrightarrow{a'} T'\}_{a'}$, and there is an injective map $\iota : \mathcal{A}' \to \mathcal{A}$ such that $T'_a \subseteq T_{\iota(a)}$ and $S'_a \supseteq S_{\iota(a)}$ for all $a \in \mathcal{A}'$, then $\mathcal{A}'$ is a *weakening* of $\mathcal{A}$ (written $\mathcal{A} \rightsquigarrow \mathcal{A}'$). $\quad\square$

**Proposition 13.** *If $\mathcal{A} \rightsquigarrow \mathcal{A}'$ and $\mu \models \Diamond\mathcal{A}$, then $\mu \models \Diamond\mathcal{A}'$.*

Proposition 13 is strictly stronger than the simple monotonicity mentioned at the beginning of the section, because a hyperarc with no targets is vacuous, so removing all targets of a hyperarc is equivalent to deleting it. It also explains why BNs and MRFs are arguably *anti*-monotonic: adding $X \to Y$ to a graph $G$ means adding $X$ to the *sources* the hyperarc whose target is $Y$, in $\mathcal{A}_G$.

As mentioned in the main body of the paper, the far more important consequence of this result is that it helps us begin to understand what QIM-compatibility means for cyclic hypergraphs. For the reader's convenience, we now restate the examples in the main text, which are really about monotonicity..

**Example 3.** Every $\mu(X, Y)$ is compatible with $\boxed{X}\rightleftarrows\boxed{Y}$. This is because this cycle is weaker than a hypergraph that can already represent any distribution, i.e., $\to\boxed{X}\to\boxed{Y} \quad \rightsquigarrow \quad \boxed{X}\rightleftarrows\boxed{Y}$. $\quad\triangle$.

**Example 4.** What $\mu(X, Y, Z)$ are compatible with the 3-cycle shown, on the right? By monotonicity, among them must be all distributions consistent with a linear chain $\to X \to Y \to Z$. Thus, any distribution in which two variables are conditionally independent given the third is compatible with the 3-cycle. Are there any distributions that are *not* compatible with this hypergraph? It is not obvious. We return to this in Section 4. $\triangle$

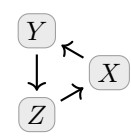

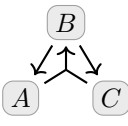

Because QIM-compatibility applies to cyclic structures, one might wonder if it also captures the independencies of undirected models. Undirected edges $A-B$ are commonly identified with a (cylic) pair of directed edges $\{A\to B, B\to A\}$, as we have implicitly done in defining $\mathcal{A}_G$. In this way, undirected graphs, too, naturally correspond to directed hypergraphs. For example, $G = A-B-C$ corresponds to the hypergraph $\mathcal{A}_G$ shown on the left. Compatibility with $\mathcal{A}_G$, however, does not coincide with any of the standard Markov properties corresponding to $G$ [13]. This may appear to be a flaw in Definition 2 (QIM-compatibility), but it is unavoidable. While both BNs and MRFs are monotonic, it is impossible to capture both classes with a monotonic definition.

**Theorem 14.** *It is possible to define a relation $\models^{\bullet}$ between distributions $\mu$ and directed hypergraphs $\mathcal{A}$ satisfying any two, but not all three, of the following.*

(monotonicity) *If $\mu \models^{\bullet} \mathcal{A}$ and $\mathcal{A} \rightsquigarrow \mathcal{A}'$, then $\mu \models^{\bullet} \mathcal{A}'$.*

(positive BN capture) *If $\mu$ satisfies the independencies $\mathcal{I}(G)$ of a dag $G$, then $\mu \models^{\bullet} \mathcal{A}_G$.*

(negative MRF capture) *If $\mu \models^{\bullet} \mathcal{A}_G$ for an undirected directed graph $G$, then $\mu$ has one of the Markov properties with respect to $G$.*

The proof is a direct and easy-to-visualize application of monotonicity (Proposition 13). Assume montonicity and positive BN capture. Let $\mu_{xor}(A, B, C)$ be the joint distribution in which $A$ and $C$ are independent fair coins, and $B = A \oplus C$ is their parity. We then have:

$$\mu_{xor} \models \quad \downarrow\overset{B}{\underset{A \quad C}{\uparrow\uparrow}}\downarrow \quad \rightsquigarrow \quad \overset{B}{\underset{A \quad C}{\swarrow\uparrow\searrow}} \quad = \quad \mathcal{A}_{A-B-C}. \quad \text{But } \mu_{xor} \not\models A \perp\!\!\!\perp C \mid B. \qquad \square$$

We emphasize that Theorem 14 has implications for the qualitative semantics of *any* graphical model (even if one were to reject the definition QIM-compatibility). We now look into the implications for some lesser-known graphical models, which may appear not to comply with Theorem 14.

**Dependecny Networks**  To readers familiar with *dependency networks (DNs)* [8], Theorem 14 may raise some conceptual issues. When $G$ is an undirected graph, $\mathcal{A}_G$ is the structure of a consistent DN.

The semantics of such a DN, which intuitively describe an independent mechanism on each hyperarc, coincide with the MRFs for $G$ (at least for positive distributions). In more detail, DN semantics are given by the fixed point of a markov chain that repeatedly generates independent samples along the hyperarcs of $\mathcal{A}_G$ for some (typically cyclic) directed graph $G$. The precise definition requires an order in which to do sampling. Although this choice doesn't matter for the "consistent DNs" that represent MRFs, it does in general. With a fixed sampling order, the DN is monotonic and captures MRFs, but can represent only BNs for which that order is a topological sort.

## C Information Theory, PDGs, and QIM-Compatibility

### C.1 More Detailed Primer on Information Theory

We now expand on the fundemental information quantities introduced at the beginning of Section 4. Let $\mu$ be a probability distribution, and be $X, Y, Z$ be (sets of) discrete random variables. The *entropy* of $X$ is the uncertainty in $X$, when it is distributed according to $\mu$, as measured by the number of bits of information needed (in expectation) needed to determine it, if the distribution $\mu$ is known. It is given by

$$\mathrm{H}_\mu(X) := \sum_{x \in \mathrm{V}(X)} \mu(X{=}x) \log \frac{1}{\mu(X{=}x)} \qquad = -\mathop{\mathbb{E}}_\mu[\log \mu(X)],$$

and a few very important properties; chief among them, $\mathrm{H}_\mu(X)$ is non-negative, and equal to zero iff $X$ is a constant according to $\mu$. The "joint entropy" $\mathrm{H}(X, Y)$ is just the entropy of the combined variable $(X, Y)$ whose values are pairs $(x, y)$ for $x \in \mathrm{V}(X), y \in \mathrm{V}(Y)$; this is the same as the entropy of the variable $X \cup Y$ when $X$ and $Y$ are themselves sets of variables.

The *conditional entropy* of $Y$ given $X$ measures the uncertainty present in $Y$ if one knows the value of $X$ (think: the information in $Y$ but not $X$), and is equivalently defined as any of the following three quantities:

$$\mathrm{H}_\mu(Y|X) := \mathop{\mathbb{E}}_\mu[\, \log {}^1\!/_{\mu(Y|X)} \,] \quad = \mathrm{H}_\mu(X, Y) - \mathrm{H}_\mu(X) \quad = \mathop{\mathbb{E}}_{x \sim \mu(X)}[\, \mathrm{H}_{\mu|X=x}(Y) \,].$$

The *mutual information* $\mathrm{I}(X; Y)$, and its conditional variant $\mathrm{I}(X; Y|Z)$, are given, respectively, by

$$\mathrm{I}_\mu(X; Y) := \mathop{\mathbb{E}}_\mu \left[ \log \frac{\mu(X, Y)}{\mu(X)\mu(Y)} \right], \quad \text{and} \quad \mathrm{I}(X; Y|Z) := \mathop{\mathbb{E}}_\mu \left[ \log \frac{\mu(X, Y, Z)\mu(Z)}{\mu(X, Z)\mu(Y, Z)} \right].$$

The former is non-negative and equal to zero iff $\mu \models X \perp\!\!\!\perp Y$, and the latter is non-negative and equal to zero iff $\mu \models X \perp\!\!\!\perp Y \mid Z$. All of these quantities are purely "structural" or "qualitative" in the sense that they are invariant to relabelings of values, and

Just as conditional entropy can be written as a linear combination of unconditional entropies, so too can conditional mutual information be written as a linear combination of unconditional mutual informations: $\mathrm{I}(X; Y|Z) = \mathrm{I}(X; (Y, Z)) - \mathrm{I}(X; Z)$. Thus conditional quantities are easily derived from the unconditional ones. But at the same time, the unconditional versions are clearly special cases of the conditional ones; for example, $\mathrm{H}_\mu(X)$ is clearly the special case of $\mathrm{H}(X|Z)$ when $Z$ is a constant (e.g., $Z = \emptyset$). Furthermore, entropy and mutual information are also interdefinable and generated by linear combinations of one another. It is easy to verify that $\mathrm{I}_\mu(X; Y) = \mathrm{H}_\mu(X) + \mathrm{H}_\mu(Y) - \mathrm{H}(X, Y)$ and $\mathrm{I}_\mu(X; Y|Z) = \mathrm{H}_\mu(X|Z) + \mathrm{H}_\mu(Y|Z) - \mathrm{H}(X, Y|Z)$, and thus mutual information is derived from entropy. Yet on the other hand, $\mathrm{I}_\mu(Y; Y) = \mathrm{H}_\mu(Y)$ and $\mathrm{I}_\mu(Y; Y|X) = \mathrm{H}_\mu(Y|X)$—thus entropy is a special case of mutual information.

### C.2 Structural Deficiency: More Motivation, and Examples

To build intuition for *IDef*, which characterizes our bounds in Section 4, we now visualize the vector $\mathbf{v}_\mathcal{A}$ for various example hypergraphs.

- Subfigures 2a, 2b, and 2c show how adding hyperarcs makes distriutions more deterministic. When $\mathcal{A}$ is the empty hypergraph, *IDef* reduces to negative entropy, and so prefers distributions that are "maximally uncertain" (e.g., Subfigures 2a and 2d). For this empty but all distributions $\mu$ have negative $IDef_\mathcal{A}(\mu) \leq 0$. In the definition of *IDef*, each hyperarc $X \to Y$ is compiled to a "cost" $H(Y|X)$ for uncertainty in $Y$ given $X$. One can see this

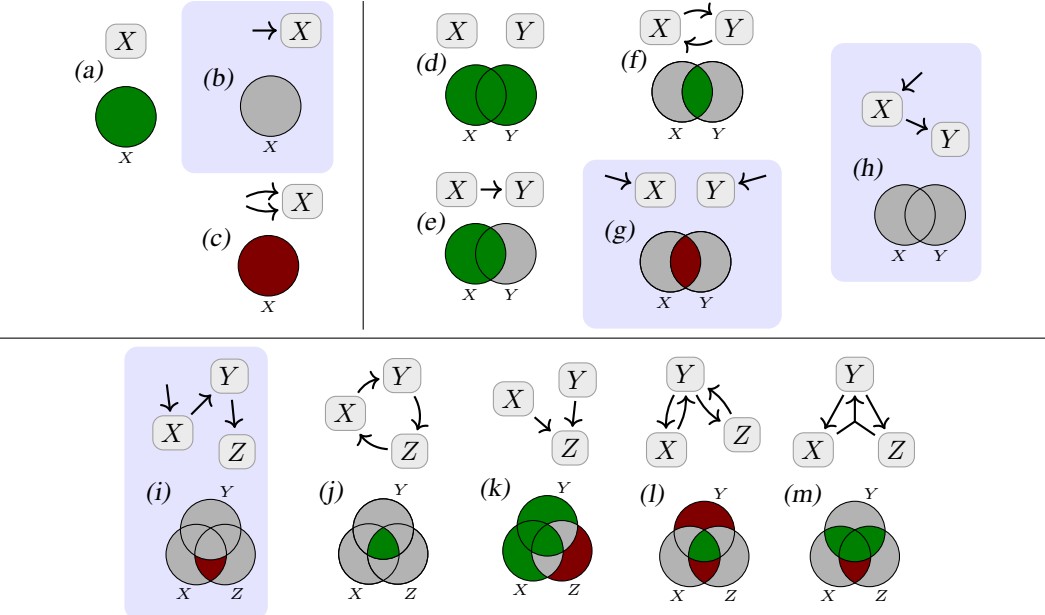

Figure 2: Illustrations of the structural deficiency $IDef_{\mathcal{A}}$ underneath drawn underneath their associated hypergraphs $\{G_i\}$. Each circle represents a variable; an area in the intersection of circles $\{C_j\}$ but outside of circles $\{D_k\}$ corresponds to information that is shared between all $C_j$'s, but not in any $D_k$. Variation of a candidate distribution $\mu$ in a green area makes its qualitative fit better (according to $IDef$), while variation in a red area makes its qualitative fit worse; grey is neutral. Only the boxed structures in blue, whose $IDef$ can be seen as measuring distance to a particular set of (conditional) independencies, are expressible as BNs.

visually in Figure 2 as a red crescent that's added to the information profile as we move from 2d to 2e to 2f.

- Some hypergraphs (see Figures 2b and 2h) are *indiscriminate*, in the sense that every distribution gets the same score (of zero, because a point mass $\delta$ always has $SDef_{\mathcal{A}}(\delta) = 0$). Such a graph has a structure such that *any* distribution can be precisely encoded by the process in (b). As shown here and also in Richardson and Halpern [23], $IDef$ can also indicate independencies and conditional independencies, illustrated respectively in Subfigures 2g and 2i.

- For more complex structures, structural information deficiency $IDef$ can represent more than independence and dependence. The cyclic structures in Examples 3 and 4, correspond to the structural deficiencies pictured in Subfigures 2f and 2j, respectively, which are functions that encourage shared information between the three variables.

### C.3    Counter-Examples to the Converse of Theorem 7

In light of Example 6 and its connections to $IDef$ through Theorem 7, one might hope this criterion is not just a bound, but a precise characterization of the distributions that are QIM-compatible with the 3-cycle. Unfortunately, it does not, and the converse of Theorem 7 is false.

**Example 7.** Suppose $\mu(X,Y,Z) = \mathrm{Unif}(X,Z)\delta\mathrm{id}(Y|X)$ and $\mathcal{A} = \{\to X, \to Y\}$, where all variables are binary. Then $IDef_{\mathcal{A}}(\mu) = 0$, but $X$ and $Y$ are not independent. $\triangle$

Here is another counter-example, of a very different kind.

**Example 8.** Suppose $A, B, C$ are binary variables. It can be shown by enumeration (see appendix) that no distribution supported on seven of the eight possible joint settings of of $\mathrm{V}(A,B,C)$ can be QIM-compatible with the 3-cycle $\mathcal{A}_{3\circ}$. Yet it is easy to find examples of such distributions $\mu$ that have positive interaction information $\mathrm{I}(A;B;C)$, and thus $IDef_{\mu}(\mathcal{A}_{3\circ}) \leq 0$ for such distributions. $\triangle$

## D   QIM-Compatibility Constructions and Counterexamples

We now give a counterexample to a simpler previously conjectured strengthening of Theorem 2, in which part (a) is an if-and-only-if. This may be surprising. In the unconditional case, it is true that, two arcs $\{\xrightarrow{1} X, \xrightarrow{2} X\}$ precisely encode that $X$ is a constant, as illustrated by Example 2. The following, slightly more general result, is an immediate correlary of Theorem 2(c).

**Proposition 15.** $\mu \models \Diamond \mathcal{A} \sqcup \{\xrightarrow{1} X, \xrightarrow{2} X\}$ *if and only if* $\mu \models \Diamond \mathcal{A}$ *and* $\mu \models \emptyset \twoheadrightarrow X$.

One can be forgiven for imagining that the conditional case would be analogous—that QIM-compatibility with a hypergraph that has two parallel arcs from $X$ to $Y$ would imply that $Y$ is a function of $X$. But this is not the case. Furthermore, our counterexample also shows that neither of the two properties we consider in the main text (requiring that $\mathcal{A}$ is partitional, or that the QIM-compatibility with $\mu$ is even) are enough to ensure this. That is, there are partitional graphs $\mathcal{A}$ such that $\mu \overset{e}{\models} \mathcal{A}$ but $\mu \not\models \Diamond \mathcal{A} \sqcup \{X \xrightarrow{1} Y, X \xrightarrow{2} Y\}$.

**Example 9.** We will construct a witness of SIM-compatibility for the hypergraph

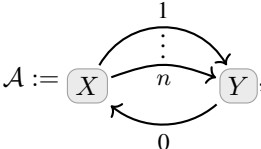

$$\mathcal{A} :=$$

in which $Y$ is *not* a function of $X$, which for $n = 3$ will disprove the analogue of Theorem 2 for the partitional context $\mathcal{A}'$ equal to the 2-cycle.

Let $\mathcal{U} = (U_0, U_1, \ldots, U_n)$ be a vector of $n$ mutually independent random coins, and $A$ is one more independent random coin. For notational convenience, define the random vector $\mathbf{U} := (U_0, \ldots, U_n)$ consisting of all variables $U_i$ except for $U_0$. Then, define variables $X$ and $Y$ according to:

$$X := (A \oplus U_1, \ldots, A \oplus U_n, \ U_0 \oplus U_1, U_0 \oplus U_2, \ldots, U_0 \oplus U_n)$$
$$= (A \oplus \mathbf{U}, \ U_0 \oplus \mathbf{U})$$
$$Y := (A, U_0 \oplus \mathbf{U}) = (A, \ U_0 \oplus U_1, U_0 \oplus U_2, \ldots, U_0 \oplus U_n),$$

where and the operation $Z \oplus \mathbf{V}$ is element-wise xor (or addition in $\mathbb{F}_2^n$), after implicitly converting the scalar $Z$ to a vector by taking $n$ copies of it. Call the resulting distribution $\nu(X, Y, \mathcal{U})$.

It we now show that $\nu$ witnesses that its marginal on $X, Y$ is QIM-compatible with $\mathcal{A}$, which is straightforward.

(b) $\mathcal{U}$ are mutually independent by assumption;

(c.0) $Y = (A, \mathbf{B})$ and $U_0$ determine $X$ according to:

$$g(A, \mathbf{B}, U_0) = (A \oplus U_0 \oplus \mathbf{B}, \ \mathbf{B})$$
$$= (A \oplus U_0 \oplus U_0 \oplus \mathbf{U}, \ U_0 \oplus \mathbf{U}) \qquad \text{since } \mathbf{B} = U_0 \oplus \mathbf{U}$$
$$= (A \oplus \mathbf{U}, U_0 \oplus \mathbf{U}) = X$$

(c.1–n) for $i \in \{1, \ldots, n\}$, $U_i$ and $X = (\mathbf{V}, \mathbf{B})$ together determine $Y$ according to

$$f_i(\mathbf{V}, \mathbf{B}, U_i) := (V_i \oplus U_i, \ \mathbf{B}) = (A \oplus U_i \oplus U_i, \ U_0 \oplus \mathbf{U}) = Y.$$

In addition, this distribution $\nu(\mathcal{U}, X, Y)$ satisfies condition

(d) $\nu(X, Y \mid \mathcal{U}) = \frac{1}{2} \mathbb{1}[g(Y, U_0) = X] \prod_{i=1}^n \mathbb{1}[f_i(X, U_i) = Y]$, since, for all joint settings of $\mathcal{U}$, there are two possible values of $(X, Y)$, corresponding to the two values of $A$, and both happen with probability $\frac{1}{2}$.

Thus, we have constructed a distribution that witnessing the fact that $\mu(X, Y) \overset{e}{\models} \mathcal{A}$.

Yet, observe that $X$ alone does not determine $Y$ in this distribution, because $X$ alone is not enough to determine $A$ (without also knowing some $U_i$).

For those who are interested, observe that the bound of Theorem 7 tells that we must satisfy

$$0 \geq IDef_{\mathcal{A}}(\mu) = - \operatorname{H}_\mu(X,Y) + n \operatorname{H}_\mu(Y \mid X) + \operatorname{H}_\mu(X \mid Y)$$
$$= - \operatorname{I}_\mu(X;Y) + (n-1) \operatorname{H}_\mu(Y \mid X)$$

Indeed, this distribution has information profile

$$\operatorname{H}(X \mid Y) = 1 \, \text{bit}, \qquad \operatorname{I}(X;Y) = n \, \text{bits}, \qquad \operatorname{H}(Y \mid X) = 1 \, \text{bit},$$

and so $IDef_{\mathcal{A}}(\mu) = -1$ bit. Intuitively, this one missing bit corresponds to the value of $A$ that is not determined by the structure of $\mathcal{A}$. $\triangle$

## E    From Causal Models to Witnesses

We now return to the "easy" direction of the correspondence between QIM-compatibility witnesses and causal models, mentioned at the beginning of Section 3.2. Given a (generalized) randomized PSEM $\mathcal{M}$, we now show that distributions $\nu \in \{\!\!\{\mathcal{M}\}\!\!\}$, are QIM-compatibility witness showing that the marginals of $\nu$ are QIM-compatible with the hypergraph $\mathcal{A}_{\mathcal{M}}$. More formally:

**Proposition 16.** *If $\mathcal{M} = (M{=}(\mathcal{U},\mathcal{V},\mathcal{F}), P)$ is a randomized PSEM, then every $\nu \in \{\!\!\{\mathcal{M}\}\!\!\}$ witnesses the QIM-compatibility of its marginal on its exogenous variables, with the dependency structure of $\mathcal{M}$. That is, for all $\nu \in \{\!\!\{\mathcal{M}\}\!\!\}$ and $\mathcal{Y} \subseteq \mathcal{U} \cup \mathrm{V}$, $\nu(\mathcal{Y}) \models \Diamond \mathcal{A}_{\mathcal{M}}$.*

The proof is straightforward: by definition, if $\nu \in \{\!\!\{\mathcal{M}\}\!\!\}$, then it must satisfy the equations, and so automatically fulfills condition (c). Condition (a) is also satisfied trivially, by assumption: the distribution we're considering is defined to be a marginal of $\nu$. Finally, (b) is also satisfied by construction: we assumed that $\mathcal{U}_{\mathcal{A}} = \{U_a\}_{a \in \mathcal{A}}$ are independent.

