# OpenReview forum: "Qualitative Mechanism Independence"
_NeurIPS.cc/2024/Conference — NeurIPS 2024 poster_

### Official Review · Reviewer_b6Hm · 2024-06-18

**Soundness:** 4
**Presentation:** 3
**Contribution:** 3
**Rating:** 7
**Confidence:** 3

**Summary:**

The paper considers the framework of directed hypergraphs and demonstrates how it can be used to represent the structural properties of probability distributions. Inspired by the successes of Bayesian networks (which are essentially DAGs), they generalize the findings from the perspective of directed hypergraphs. The key innovation seems to be that instead of simply understanding conditional dependencies (or more specific context-specific indepdencies), the framework goes beyond to consider functional dependencies and understand thesein greater depth. The idea is to use these probabilisitic dependency graphs  as formulations to define the idea of QIM (qualtiatively independent mechanism) compatibility of a distribution w.r.t a hypergraph.

Once the QIM compatibility is defined, the paper proceeds to demonstrate its usefulness in two specific settings - causal models and information theory. Specifically, it establishess the equivalence between QIM compatibility and randomzized probabilistic structural equation models (PSEMs). The key result in this direction is presented in Proposition 4 where it presents a natural generalization of causal model that exactly captures QIM compatibilitywith an arbitrary hypergraph.  In addition, the paper considers the correspondence between QIM compatibility and interventions in causal models. Finally, it takes a deeper dive into discussing the relation between information theory and QIM compatibility by defining a qualitative scoring function for probabilisitic dependency graphs. The key idea here is to show how one can measure how far a distribution is QIM compatible with a hypergraph structure.

**Strengths:**

Over all, this is a well written paper. Albeit quite dense, the paper is indeed worth publishing for the community. It was quite insightful read and the paper clearly layes out the problem and the solutions. Clearly written for a niche community, the paper does convey what it aims to do -- the value of probabilistic hyerpgraphs in deeply understanding the causal models and the connection to information theory. I q2uite like the paper.

**Weaknesses:**

While the paper is excellent, I do have a few concerns/questions:

* I am not sure I clearly see what the contributions of the current paper are w.r.t the literature. For instance, as the paper mentions., Richardson and Halpern had already defined PDGs but you have redefined it in Definition 1. This is perfectly fine if this section is a background section but the way it is written, it appears that it is part of the contributions.  It would be great to separate out where teh prior work and background ends and where the paper begins. I assumed that it ends around line 94 and the paper's contributions start from Definition 2. Is this correct?

* As the paper itself mentions, I do not see the need for Proposition 3. It kinda directly follows from Theorem 1 and so why have it separately when the value is not clear?

* While I clearly like the information theory part more (since the scoring function is quite intuitive at the end due to the definition of QIM compatibility), I would have liked to see some practical use cases. I would have liked to see a specific discussion on the types of settings/problems where such situations are plausible/common. Specifically, what kind of interventions are possible (may be in specific domains).

* Some analysis on the computational complexity of these scoring functions would be nice as well.

Over all, a good paper but can be made a bit more acecssible with some real examples.

**Questions:**

Please see the weaknesses for specific questions about theorems, definitions and the delination between prior and proposed contributions.

**Limitations:**

Some real examples could be used for motivation.

---

> ### Author Rebuttal · Authors · 2024-08-07
>
> Thank you for your review!
> A few responses to your questions:
>
> 1. **The division between hypergraphs (defn 1) and original contributions.** Definition 1 is an old (if not particularly common) idea; we cite Gallo et. al. [4] as a reference, although surely they were not the first to study directed hypergraphs. In any case, Definition 1 is not the definition of a PDG.  It is important to distinguish directed hypergraphs from PDGs for the same reason that it is important to distinguish ordinary directed acyclic graphs (dags) from Bayesian Networks.  We present a definition of a directed hypergraph for two reasons: to make the paper self-contained, and to introduce convenient notation.
>
>     You are essentially correct about where the division between prior work and the contributions are, although arguably the contributions start with the lead-in to Definition 2, on line 86. We will try to add a few words to try to make it clearer where the background ends and our contributions begin.
>
> 2. **Distinguishing Proposition 3 from Theorem 1.**
> Proposition 3 is in many ways orthogonal to Theorem 1.  Theorem 1 describes an equivalence between the independencies of a dag $G$ and QIM-compatibility with $\mathcal A_G$, but says nothing about causality.  Meanwhile, Proposition 3 describes an equivalence between arising from a fully randomized causal model with graph $G$ (now an arbitrary graph, not necessarily acyclic), and QIM-compatibility with the graph $G$.  By composing the two results, we get an equivalence between BNs  and (fully randomized) acyclic causal models.  But Proposition 3 does not follow from Theorem 1, because Theorem 1 does not talk about causality (or graphs with cycles, for that matter).
>
>
> 3. **Practical use cases.**
> You are right to point out that our examples are often purely mathematical and uninterpreted. We will think hard about how to enrich our examples to more vividly illustrate the utility of our approach to qualitative modeling.
>
>     We are not entirely sure what is meant by the inquiry as to “what interventions are possible”. This work is focused on qualitative modeling, while interventions typically involve concrete values of the variables, and one is typically interested in the quantitative effects of those interventions. Nevertheless, as we show in Theorem 7, a witness to qualitative compatibility can be used to describe interventions. Although the statement of the theorem describes a limitation as to which interventions are encoded in a witness (a certain event must have positive probability), we believe that those limitations are surmountable (see footnote 2).  We also suspect that a generalized randomized PSEM (definition 4) may be able to handle a far broader class of interventions than the standard one—but investigating that is beyond the scope of this work.
>
> 4. **Complexity.** Because there are $2^n$ joint distributions over $n$ variables, even the input to the scoring functions must be exponentially large, unless we restrict to a specialized structure parameterizing a special class of distributions (that have certain (in)dependencies). In any standard representation, the difficulty of calculating IDef is dominated by the difficulty of calculating the entropy of the entire joint distribution.  While difficult in some cases (e.g., certain undirected models), often it is the case (e.g., in Bayesian Networks or clique trees) that the same independence structure that makes for a manageable representation also makes it easy to calculate the entropy of the distribution it represents.
>
>     Calculating QIM-Inc, on the other hand, is significantly more difficult. (In fact, even calculating whether or not a distribution is QIM-compatible with a hypergraph appears to be quite difficult.) It involves solving an optimization problem over extended distributions—objects that are (perhaps even exponentially) larger even than the original distributions.  And even if we ignore the cost of representing extended distributions, how long will it take to solve the optimization problem?  This is an interesting question, and one we do not yet have a good theoretical understanding of.  We do have a rudimentary approach that is able to solve that problem in practice, providing a witness of QIM-compatibility if there is one. But it works only for small graphs, and we have not yet been able to prove its correctness. These problems remain important areas for future research.

---

> > ### Comment · Reviewer_b6Hm · 2024-08-09
> > **Thanks for the response**
> >
> > Thank you for the detailed response to my questions. Almost all my concerns are addressed in your rebuttal. Sincerely appreciatre your time in responding.
> >
> > I would suggest that you please include the complexity discussion that you have presented here and real examples in the next iteration of the paper. These will only enhance the paper.

---

### Official Review · Reviewer_3P8H · 2024-07-11

**Soundness:** 2
**Presentation:** 1
**Contribution:** 1
**Rating:** 4
**Confidence:** 4

**Summary:**

The paper presents a formalism that is claimed to extend the qualitative structure of probabilistic dependency graphs.

**Strengths:**

I think it might be original, but it is difficult to tell. It might be potentially significant, but it isn't clear what the significance might be.

**Weaknesses:**

It is not clear what problem this solves (or does better than other proposals). It claims to be "do much more", but it isn't clear what the much more is. The semantics needs to be presented in a much more straightforward way.

In example 4, you ask "are there distributions not compatible...? It is not obvious." The answer is yes. The parity function X \equiv (Y \equiv Z) that is true when an even number of X,Y,Z are true, is not compatible. They are each a function of the other two but are independent of either one.

**Questions:**

What is the problem that this is a solution to? What can this do (or do better) that other proposals cannot do?

You highlight that this is a hypergraph rather than a DAG in a belief network. What does the hypergraph let us do that a Bayesian network does not? (In a Bayesian network, all of the parents affect the child; your's allows multiple targets.) Can a node be in multiple targets?  If so, what if different mechanisms result in different values? If not, isn't having a mechanism that produces multiple targets trivially equivalent to multiple mechanisms that produce single targets?

For Example 3 with Boolean variables, the bidirectional arrow (X <-> Y) has 4 parameters, but there are only 3 degrees of freedom. Almost all parameterizations are inconsistent. For one of the inconsistent parameterizations, does it define a distribution? If so, which one?

**Limitations:**

I don't think it has any societal impacts, positive or negative.

There are very few limitations to the work done.

---

> ### Author Rebuttal · Authors · 2024-08-07
>
> Thank you for your review!
>
> **QUESTIONS**
>
> 1. You start with a very important question:
> > what is the problem that this is a solution to? What can it do (or do better) that other proposals cannot do?”
>
>     Until now, there has not been a satisfying generalization of qualitative Bayesian networks (which can capture independencies without needing conditional probability tables) that applies more broadly to other (causal/probabilistic) models, such as those with cycles or constraints. At the same time, there has been no definition for what it would mean for a distribution to be compatible with an arbitrary set of causal mechanisms.  The answer we propose in this paper not only solves these problems, but also provides a deep and nontrivial connection between information theory and causality.  For the causality community, it provides an information-theoretic test for whether a probability distribution is compatible with a (possibly cyclic) qualitative graph.  For the information-theory community, it provides a principled notion of “pairwise interaction” that explains and shines light on a long misunderstood quantity (interaction information) and reframes a standard counterexample in new light.
>
> 2. Next, for your questions about expressiveness, and the syntax of directed hypergraphs.
> > What does the hypergraph let us do that a BN does not? (In a BN, all parents affect the child; your's allows multiple targets.) Can a node be in multiple targets? If so, what if different mechanisms result in different values?
>
>     (Directed) hypergraphs can express many things that DAGs cannot.  They can encode cyclic graphs.  They can also describe situations in which multiple independent mechanisms generate the same variable, as shown in Example 2  (so a node can indeed be in multiple targets). In such situations, all mechanisms that generate a node are required to generate the same value. This may seem strange, but it is useful, and there are scenarios where it is also quite natural. Suppose that X and Y turn out to be different names for essentially the same concept. Then, in addition to the ordinary mechanism that explains the value of X, we can also add an equation stating that $X = f(Y)$ for some function f, which must also hold simultaneously. In such cases, the question of “which value of X: the one prescribed by the original function, or the one prescribed by Y?” is moot; both equations must hold.
>
>     Finally, you ask if having a single mechanism that produces, say, three targets (call them $X,Y,Z$) is the same as having three (independent) mechanisms that each produce a single target ($X,Y$, and $Z$, respectively).  They are not, because the mechanisms are assumed to be *independent*.  A single randomized function that produces all three of $(X,Y,Z)$ can represent an arbitrary distribution over the three variables, but three independent randomized functions taken together can only represent distributions of the form $P(X,Y,Z) = P(X) P(Y) P(Z)$.
>
> 3. We are not sure that we understood your final question, but we will try to answer it to the best of our ability.
> > For Example 3 with Boolean variables, the bidirectional arrow (X <-> Y) has 4 parameters, but there are only 3 degrees of freedom. Almost all parameterizations are inconsistent. For an inconsistent parameterization, does it define a distribution? If so, which one?
>     By the “bidirectional arrow X <-> Y”,  we assume you mean the pair of two arrows, X->Y and Y->X.  We have not talked in this paper about how to parameterize arrows (we are focused on the qualitative aspects of the graph, not the probabilistic parameterization).  However, the work on probabilistic dependency graphs (PDGs) that we reference does.
>
>     In that setting, parameters for these two arrows amount to giving four distributions:  P(Y|X=0) and P(Y|X=1) for the arrow X -> Y, as well as P(X|Y=0) and P(X|Y=1) for the arrow Y-> X. Each is a distribution over a binary variable, and hence can be specified with a single parameter.  So, if we have understood correctly, this indeed means specifying four independent parameters, when in fact there are only three degrees of freedom in joint distributions over P(X,Y).  As you correctly point out, this means most settings of the parameters will not be consistent with any distribution. The question of which distribution is specified in this case is an interesting one; indeed, that is the focus of Richardson and Halpern’s 2021 paper on PDGs. (Essentially, they do so by finding a distribution that is minimally inconsistent with the specified information, as measured by relative entropy.)  But that resolution of the inconsistency is not directly relevant to our paper.
>
> **WEAKNESSES**
>
> We admit that the presentation is dense and technical; we have tried to add intuition and examples to mitigate this, and may be able to do more with an extra page. We welcome any suggestions for how to make the presentation more straightforward without making it less precise.
>
> You provide an answer to the rhetorical question posed in example 4—and indeed, when we return to the example in section 4 (as promised in the forward reference), we give exactly the same answer (example 5).  We agree with the reviewer that this distribution ($\mu_{\mathrm{xor}}$ in the paper) intuitively should not be compatible with the 3-cycle. But why not?  Can you prove it without using the tools presented in Section 4 (Theorems 7 or 8)?  We found this exercise very difficult, and were able to do it only by combining a technical argument tailored to binary variables with exhaustive computer search.  We are eager to hear if you have found a simpler proof, especially one that does not use the information-theoretic arguments behind Theorem 7! (See Example 6 for further evidence that questions of QIM-compatibility may not always be as obvious as they may seem.)

---

> > ### Author Response · Authors · 2024-08-08
> > **(missing line break and mis-scoped quotation)**
> >
> > We just noticed a formatting error in our response, and would like to head off any confusion; our response to the third question begins inside the quoted material, due to a missing line break.  Explicitly, our response to your third question should instead begin as follows:
> >
> > ---
> >
> > We are not sure that we understood your final question, but we will try to answer it to the best of our ability.
> >
> > > For Example 3 with Boolean variables, the bidirectional arrow (X <-> Y) has 4 parameters, but there are only 3 degrees of freedom. Almost all parameterizations are inconsistent. For an inconsistent parameterization, does it define a distribution? If so, which one?
> >
> > By the “bidirectional arrow X <-> Y”, we assume you mean the pair of two arrows, X->Y and Y->X. We have not talked in this paper about how to parameterize arrows (we are focused on the qualitative aspects of the graph, not the probabilistic parameterization). However, the work on probabilistic dependency graphs (PDGs) that we reference does.
> >
> > In that setting ...
> >
> > ---
> >
> > Apologies for the oversight!

---

> ### Comment · Reviewer_3P8H · 2024-08-12
> **reply**
>
> There are many generalizarions of directed causal networks to include constraints (see e.g., the books and papers of Rina Dechter) and cycles (typically taken as the equilibrium distribution of a Markov chain; for example https://rss.onlinelibrary.wiley.com/doi/abs/10.1111/1467-9868.00340, https://jmlr.csail.mit.edu/papers/volume1/heckerman00a/heckerman00a.pdf, https://www.ijcai.org/Proceedings/13/Papers/161.pdf).
>
> I thought the parity example was an obvious counter-example; because the variable was independent of each of the other variables, that it was clear. One theoretical justification is in terms of the Hadamard transform (or the discrete Fourier); one reference where this is applied to graphical models is https://proceedings.mlr.press/v22/buchman12.html  The cycle loses the high frequency terms (the one needed for the parity term).

---

> > ### Author Response · Authors · 2024-08-12
> >
> > > There are many generalizarions of directed causal networks to include constraints (see e.g., the books and papers of Rina Dechter) and cycles (typically taken as the equilibrium distribution of a Markov chain; for example https://rss.onlinelibrary.wiley.com/doi/abs/10.1111/1467-9868.00340, https://jmlr.csail.mit.edu/papers/volume1/heckerman00a/heckerman00a.pdf, https://www.ijcai.org/Proceedings/13/Papers/161.pdf).
> >
> > It is true that many have considered generalizations of directed causal networks to include cycles and constraints, and we are aware of the references you point out. (In fact, we cut a discussion of Heckerman's dependency networks to streamline the story.)  Yet none of these papers provide a satisfying answer to what these models mean at a qualitative level. (Heckerman shows that "consistent" DNs capture the same distributions as undirected graphical models, but this characterization only applies to special structures.)  For acyclic models, there is an obvious answer: the structure implies certain independencies.  But what is the analogue for a cyclic model?  What can you say about the stationary distribution of a Markov chain? To answer this question you have to be more precise about what Markov chain you're talking about. In order to define an equilibrium semantics as you suggest  (or indeed, to even formally define a Markov Chain for a cyclic network in which the state is a joint distribution), it is necessary to make a structural choice, that in a sense, breaks the symmetry promised by a cyclic representation.  This choice can be made in the form of a sampling order (as is an important point in the Heckerman (2000) and Poole&Crowley (2013) papers you reference), or a cut set (as in the Baier et. al. (2022) paper that we reference).  Either choice amounts to a selection of qualitative information that is not present in the underlying graph, and often swept under the rug. It is not hard to show that a choice of sampling order actually induces a *BN's* independencies, and therefore this approach does not say anything interesting about cyclic models at a qualitative level.
> >
> > We also point out that Poole and Crowley paper you reference states that there "seem to be three solutions to causal modeling with cycles: (1) do not allow cycles, (2) make noise dependent, or (3) use a different (non-causal) semantics".  Yet our approach uses causal semantics with independent noise, and allows for cycles!
> >
> > > I thought the parity example was an obvious counter-example; because the variable was independent of each of the other variables, that it was clear. One theoretical justification is in terms of the Hadamard transform (or the discrete Fourier); one reference where this is applied to graphical models is https://proceedings.mlr.press/v22/buchman12.html The cycle loses the high frequency terms (the one needed for the parity term).
> >
> > We too found the parity distribution to be an obvious candidate for a counter-example. Yet, as mentioned in our response, we found it (surprisingly) difficult to establish that there could be no witness satisfying the properties of our definition.  While we understand that the Hadamard and discrete Fourier transforms are intimately related to parity systems, we do not see any way to apply them to demonstrate a lack of QIM-compatibility.  Your intuition that a cycle should "lose high-frequency terms" concords with ours; indeed, such an argument can be used to show that the parity distribution $\mu_{\mathrm{xor}}$ cannot be written as  $\mu_{\mathrm{xor}}(X,Y,Z) = f_1(X,Y) f_2(Y,Z) f_3(Z,X)$, for any choice of $f_1,f_2,f_3$.  Yet despite having these intuitions, we still were very surprised how difficult it was to provide a formal proof that the parity distribution is not QIM-compatible (in the sense of Definition 2) with the 3-cycle.
> >
> > Fortunately (in our opinion), the effort paid off in the general case: the information-theoretic test for QIM-compatibility with the 3-cycle is entirely novel, quite different from more standard spectral arguments (or those that rest on polynomial degrees), and has also helped to clarify the meaning of interaction information.

---

### Official Review · Reviewer_wE8z · 2024-07-19

**Soundness:** 3
**Presentation:** 2
**Contribution:** 3
**Rating:** 7
**Confidence:** 4

**Summary:**

The paper studies notions of "compatibility" between probability distributions and directed hypergraphs with causal mechanisms.  In such a hypergraph, we have (roughly) hyperedges T ---> S annotated by a latent/exogenous variable U where the variables S are functionally determined by the variables T and U.  Bayesian networks can be naturally formulated in such terms (where T is the set of parents and S is a singleton set containing the child, and U is a (local) causal mechanism).  The paper explores notions of "compatibility" with a joint distribution (over say endogenous variables) and such a hypergraph with additional causal mechanisms.  "Compatibility" here means that there exists an realization of the hypergraph that matches a given distribution.

The paper shows that this notion of compatibility can:
1) characterize the conditional independencies of a distribution and a DAG
2) based on hypergraph structure, can represent some functional dependencies (unlike DAGs)
3) characterize (probabilistic) structural equation models (SEMs) for a graph
4) characterize generalized (probabilistic) SEMS for hypergraphs
5) given a (witness) distribution for a class of hypergraphs, can reconstruct its PSEM (up to a family of PSEMs)
6) and, from (5), also characterize its interventional distrbution
7) compatibility implies a negative "information deficiency"
8) be characterized with a QIM incompatibility score based on information theoretic entropy
9) which, from (8), can also be upper- and lower-bounded

(note: my summary may be over-simplifying).  Correspondingly, notions of independence, causality and information theory are tied together through the notion of compatibility.

**Strengths:**

The paper is ambitious, and seeks to tie together central ideas from independence, causality, and information theory into a notion of distribution compatibility with a hypergraph over causal mechanisms.  In addition, there are some generalizations made from (causal) DAGs and SEMs to hypergraphs.

I appreciate that the authors regularly included examples throughout the paper, which makes things easier to follow, and also helps to motivate the discussion.

**Weaknesses:**

At times, the claims of the paper seem overstated.  The first half of the results are familiar from (causal) DAGs.  For example, the claim after Proposition 3, mentions a phenomena relating randomized SEMs and BNs as possibly not being formalized previously.  I believe, for one example, that the following paper talks about this connection:

  "Causality in Bayesian belief networks"
  Marek Druzdzel, Herbert Simon
  in Uncertainty in Artificial Intelligence, 1993.

I also found the paper relatively dense, with many results given.  I believe there is a central theme of compatibility tying together concepts of independence, causality and information theory.  Some of the other results, for example, the characeterization of some functional dependencies through hypergraph edges --- this seems more like an extra result that does not clearly support the main story (in my opinion), or it is otherwise under-explored given the space constraints.

**Questions:**

none

---

> ### Author Rebuttal · Authors · 2024-08-07
>
> Thank you for the reference to Druzdzel and Simon’s 1993 paper!  You are right to point out a similarity between their Theorem 1 and ours; we both provide causal interpretations of Bayesian Networks. We will certainly discuss this in the full paper! However, upon close examination, we believe that their results are not as strong as ours (i.e., do not fully capture the equivalence between BNs and fully randomized acyclic causal models). Here is why:
>
> - At best, Druzdzel and Simon’s Theorems 1 (and 2) provide one direction of the result: if you have a Bayesian Network (with graphical structure G) then it can be converted to an equivalent structural equations model (also with structure G) with independent error terms.  But the reverse direction is missing: that an arbitrary structural equations model with independent error variables determines a Bayesian Network.
> - While it is true that Druzdzel and Simon’s construction could be the centerpiece of an alternate proof of one direction of our Theorem 1, this fact is not reflected in the theorem statement, which says nothing about the mechanisms’ noise variables being (jointly) independent.  Thus, technically speaking, the formal statement of their result is weaker than (one direction of) ours. Indeed, strengthening the statement is necessary to get the reverse direction of the equivalence.
> - Perhaps most importantly, the fundamental story is extremely different. Druzdzel and Simon are concerned with interpreting *quantitative* BNs in a causal way (and to that end finish with a third theorem that is unrelated to our work). But they are not concerned with representing independencies in this way. Indeed, they state in the intro  that “a causal structure does not necessarily imply independences”, strongly suggesting that they did not realize that their result could actually form half of a causal characterization of BN independencies.
>
> We share your intuition that this equivalence between (fully randomized) causal models and Bayesian networks should be present in the literature. Beyond the Druzdzel and Simon reference that you pointed us to, which comes close to capturing one direction of the correspondence, we also recently (re)discovered that Pearl’s “Causal Markov Condition” (Theorem 1 from Causal Inference: an overview, 2009) comes close to showing the other direction.  Yet we still haven’t found anything that puts the two halves together and recognizes it as an equivalent characterization of a Bayesian Network’s conditional independencies. We will make sure to point out these new points of contact with the literature, and will tone down the rhetoric accordingly. We would also be grateful for any other leads you might have!
>
>
> Now, to directly address your concerns:
>
> **Overstatement.** Whether or not we can track down prior discussion of the equivalence discussed above, we are happy to include a discussion of how Pearl's and Druzdel and Simon’s work relates to ours, which will mean toning down the rhetoric that follows Proposition 3.  Please let us know if there are other places in which the results seemed overstated!
>
> **Density and “Extra” Results.** We agree that the paper is denser than would be ideal, but found that this level of density was necessary to capture the full scope of the concept. Theorem 2 is identified as one result that might not directly support the main story.  But, to our minds, Theorem 2 is a key element.  It demonstrates that mechanism independence can capture not only (conditional) independencies but also functional dependencies, even within a single hypergraph. That our definition can capture both dependence and independence is no coincidence—the two are closely related (see lines 28-31) and both are special cases of the information-theoretic constraints that QIM compatibility implies in general (see lines 313-315, or for more detail, lines 333-335, equation (2), and Theorem 7). If the paper is accepted, perhaps with the extra page provided in the proceedings we will be able to make this thread more visible, and otherwise add discussion to reduce the density of the paper.

---

### Official Review · Reviewer_91DJ · 2024-07-19

**Soundness:** 3
**Presentation:** 2
**Contribution:** 3
**Rating:** 7
**Confidence:** 2

**Summary:**

This paper establishes a notion of "QIM compatibility" between the functional dependences and the joint distribution through the directed hypergraph. The functional dependence is a general notion of dependences containing conditional independences.

**Strengths:**

I want to state that my understanding of this paper is partial, given that I have a limited knowledge and understanding on the computational logic theory where this paper might reside in. I am from causal inference field.

---
1. This paper is technically precise. All mathematical terminologies are carefully chosen, and the degree of ambiguity is minimized.
2. I think this theory has a lot of potentials in providing a graphical tool for describing functional dependences. Despite the wide usage of the causal graph, it's known that the graph is only suitable for expressing conditional independences. Even if the causal graph also carries functional independences (called "Verma's constraints" described exemplified in Question section), such constraints are not explicitly shown in the graph. I think the proposed framework has a potential of explicitly revealing such hidden constraints from the graph.

**Weaknesses:**

I want to state that my understanding of this paper is partial, given that I have a limited knowledge and understanding on the computational logic theory where this paper might reside in. I am from causal inference field.

---

__Lack of preliminaries__

I felt difficulty in digesting this paper.  Some knowledge on probabilistic dependency graph (PDG) is required to understand this paper, but examples are limited to capture what functional independences are captured from this PDG. Also, a natural question is then the difference between the DAG and the PDG. Such differences need to be highlighted to motivate this work and provide a clear distinction on the notion of PDG.

__Lack of motivations__

I think a real-world example of functional independences that are not captured by the conditional independence terminology is required. Even if there are some examples (such as a non-random coin in line 134) exist to capture the distinction between the causal graph and the PDG, this example is somewhat made-up and can be addressed in the existing framework, since in causal graph, such two non-random coins are considered as one variable. I believe this proposed work is providing a new _paradigm_ compared to existing causal graphical model to capture more general functional dependencies. Then, there should be more examples that readers would feel the incapability of causal graphs.


__Weak literature review__
I wanted to read the history of development of the notion of QIM in the paper but couldn't find. What is the limitation of previous papers, and what are the distinction of this paper compared to them?

**Questions:**

1. My understanding of QIM compatibility from Definition 2 is that the hypergraph $\mathcal{A}$ is QIM-compatible if $\mathcal{A}$ satisfies the causal Markov condition with respect to the distribution. Then, there must be a graph capturing this independence information, by a graphoid theory. Then, how the notion of QIM compatibility can be differentiated with the existing graphoid theory?

2. A semi-Markovian causal graph (an acyclic directed mixed graph, ADMG) is known to carry a set of conditional independences and a set  of _functional independence_. For example, consider a graph G = {W -> R -> X -> Y, W<-> X, W<-> Y} (which is known to be a _Napkin graph_ (Book of Why, Pearl)). In this graph, no conditional independences exist. However, it's known that the functional $Q[Y]:= \frac{\sum_{w}P(y,x \mid r,w)P(w)}{\sum_{w}P(x \mid r,w)P(w) }$ (which is an identification estimand of $P(y | do(x))$) is known to be independent of the choice of $r$. This type of functional independence is called the _Verma's constrains_. This type of constraints doesn't show up explicitly in the ADMG. Then, do you think this type of constraints (more generally, a set of conditional independence and Verma's constraints) can be shown simultaneously in the directed hypergraph?

**Limitations:**

1. I think this paper only considered the case where the unmeasured noises are independent. I think this is a strong assumption.

---

> ### Author Rebuttal · Authors · 2024-08-07
>
> Thank you for your review!
>
> **RESPONSES TO QUESTIONS**
>
> 1. While your understanding is not far off at a high level, there is an important wrinkle in your restatement of Definition 2: what exactly do you mean by “the causal Markov condition (for a hypergraph) with respect to a distribution?”  There is no standard answer to this question. In a way, the primary contribution of this paper is to propose a (novel) principled way of making this precise.  Yet to do so, we have had to step beyond the purview of graphoid theory.  As mentioned in the introduction, the standard theory of graphoids cannot even describe functional dependencies, let alone the complex information-theoretic constraints implied by QIM compatibility in general (e.g., Theorem 7). We reiterate that QIM-compatibility is about more than just independence; for a very concrete illustration, see Example 2.
>
> 2. This is an interesting idea. To model an ADMG with a directed hypergraph, it seems the appropriate thing to do would be to explicitly include the implied confounding variables.  Our (brief) investigation has not yet turned up anything interesting about these Verma’s constraints, and it is not yet clear to us what relationship they may have to the hypergraph.  But going forwards, we will keep an eye out for these properties. Thanks again for the suggestion!
>
> **ADDRESSING WEAKNESSES AND LIMITATIONS**
>
> **Preliminaries.** QIM compatibility is an original and self-contained concept. Although our concept of QIM compatibility turns out to have subtle connections to existing work on probabilistic dependency graphs (PDGs), as we show in section 4, we maintain that no prior knowledge of PDGs is necessary to understand any part of this paper.  We hope that some careful rewording of the paragraphs in the introduction and in section 4 will clarify this.
>
> **Motivations.** The reviewer makes a good point: many of our examples (such as the non-random coin on example on line 134) can be captured with the existing framework for causal modeling. Recall, however, that our goal is to develop a unified graphical language for just the qualitative aspects of causality (such as independence and dependence). In that regard, Example 2 demonstrates a lack of expressive power of qualitative (causal) Bayesian Networks: they cannot represent determinism.
>
> Indeed, there are two standard ways that one can interpret a graph as specifying the qualitative structure of a causal model. Either (1) each variable $X$ is associated with a function $f_X(\mathrm{Pa}(X))$ that determines the value of $X$ based on the value of its parent variables, or (2) the function $f_X$ also takes as input an additional independent noise (this is Pearl’s definition; we call it a randomized causal model). Quantitatively, the two are equally expressive, because noise can be modeled explicitly as variables, and, conversely, one can ignore the noise term.  But qualitatively, these two standard models are different, and neither can express both dependence and independence. Graphs with interpretation (1) cannot articulate independencies, while graphs with interpretation (2) cannot describe dependencies or determinism. Our framework can do both, even within a single model. It can also capture subtler qualitative phenomena, as we explore in Section 4.
>
> **Literature Review.** As far as we are aware, the definition of qualitatively independent mechanisms (QIM), in the general form made precise by Definition 2, is entirely new.  Of course, people have long studied the special case of hypergraphs that arise from DAGs (which, by Theorem 1, amounts to Bayesian Network independencies); we have credited Pearl (Causality, 2009) for the intuition for DAGs, on which our more general notion is based.  The other key point of contact with the literature is probabilistic dependency graphs (PDGs).  The key difference between the hypergraphs we work with here and PDGs is in the semantics they give to qualitative information (and the fact that the hypergraphs don’t deal with quantitative information at all, whereas PDGs do).  PDGs give semantics to qualitative information using an information-theoretic scoring function (IDef); we do so using QIM-compatibility (Definition 2).   But, as we show (Theorems 7 and 8), there are deep connections between the two.
>
> **Independence of Noise Variables.** It is indeed a strong assumption to assume that noise variables are independent, but it is one that is commonly made both in theory (see Pearl’s definition of a causal model in Causality (2009)) and in practice.  Moreover, perhaps counter-intuitively, it is still possible to use this framework to express situations in which noise variables are not independent.  In fact, it is often possible to do so in two different ways: one can either (1) combine mechanisms that are not independent into one large mechanism with the union of the sources and targets, or (2) explicitly model the noise as variables on which both mechanisms can depend.  As mentioned in the paper, randomized causal models (which assume independent noise) and ordinary causal models (which do not) are equally expressive.

---

> > ### Comment · Reviewer_91DJ · 2024-08-10
> > **Response**
> >
> > Thank you for carefully addressing my questions and concerns. I believe QIM has the potential to provide additional independence information that cannot be captured by a graph. Based on this, I will raise my score.

---

### Author Rebuttal · Authors · 2024-08-07

Thank you all for your careful reading and useful comments!

First and foremost, we want to emphasize that our work has focused purely on qualitative aspects of a model (those properties that can be described with a graphical structure, without needing to know, for instance, the specific values that variables can take on). Indeed, the central definition of the paper (Definition 2) does not discuss how we can use (directed) hypergraphs as the basis of a quantitative modeling tool. Therefore, our framework cannot be used to model specific concrete distribution at all (although it can certainly be used to characterize structural properties of a concrete distribution). To model concrete distributions, one would need to augment directed hypergraphs with additional information, in the same way that one augments a causal graph with equations to get a SEM, or augments a directed acyclic graph with probability tables to get a Bayesian Network.  Correspondingly, we mention two possible avenues for augmenting (directed) hypergraphs with quantitative information: we can annotate hyperarcs with (randomized) functions, to get Generalized Randomized PSEMs (Definition 4), or we can annotate them with (possibly weighted) probability tables, to get a PDG [Richardson and Halpern, 2021].  In Sections 3 and 4, respectively, we describe some relationships between QIM-compatibility and each of these two models. Nevertheless, ultimately our work here is at the qualitative level.  Our goal was to show that our formalism could capture in a useful way qualitative phenomena such as dependence, independence, and more, with QIM compatibility. We believe we have shown that we can.

Perhaps in part due to our focus on unifying qualitative models, the examples we have chosen are not concrete real-world situations in which a modeler should be using our framework; they were instead selected to be mathematically simple and illustrate key conceptual points about qualitative modeling.  Having said that, we will think hard about how to modify our examples to more forcefully make the case that this level of generality is truly necessary to represent what one would like, at the qualitative level.   We agree that this would give the paper more impact, and appreciate the suggestion.

Several reviewers point out that the material is rather dense. Indeed it is, and if the paper is accepted, we will do what we can with the additional page in the proceedings to expand on and clarify the material that is already there.

---

### Decision · Program_Chairs · 2024-09-25

**Decision:**

Accept (poster)

**Comment:**

The paper studies notions of “Compatibility”  between probability distributions and directed hypergraphs. Its main benefit beyond Bayesian networks is that it can capture functional dependencies.
The majority of f the reviewers agreed that the paper can be accepted even when they were not sure about the contribution beyond earlier work.One reviewer questions the very essence of the work: what does it add to the area. What is the question being solved here. Yet after discussion he also agreed that given  all the points raised in the discussion the paper ca be accepted.